# All you need is spin: SU(2) equivariant variational quantum circuits based on spin networks

**Richard D. P. East**[⋆], **Guillermo Alonso-Linaje** and **Chae-Yeun Park**[†]

Xanadu, Toronto, ON, M5G 2C8, Canada

⋆ rdp.east@gmail.com , † chae.yeun.park@gmail.com

## Abstract

Variational algorithms require architectures that naturally constrain the optimisation space to run efficiently. In geometric quantum machine learning, one achieves this by encoding group structure into parameterised quantum circuits to include the symmetries of a problem as an inductive bias. However, constructing such circuits is challenging as a concrete guiding principle has yet to emerge. In this paper, we propose the use of *spin networks*, a form of directed tensor network invariant under a group transformation, to devise SU(2) equivariant quantum circuit ansätze – circuits possessing spin rotation symmetry. By changing to the basis that block diagonalises SU(2) group action, these networks provide a natural building block for constructing parameterised equivariant quantum circuits. We prove that our construction is mathematically equivalent to other known constructions, such as those based on twirling and generalised permutations, but more direct to implement on quantum hardware. The efficacy of our constructed circuits is tested by solving the ground state problem of SU(2) symmetric Heisenberg models on the one-dimensional triangular lattice and on the Kagome lattice. Our results highlight that our equivariant circuits boost the performance of quantum variational algorithms, indicating broader applicability to other real-world problems.

## Contents

---

# 1 Introduction

Variational algorithms are prominent across physics as well as computer science with particularly fruitful applications in machine learning, condensed matter physics, and quantum chemistry [1–4]. In such areas, a parameterized function, often called an ansatz, is used to model a probability distribution or a quantum state, and parameters are optimised by minimising a cost function. However, this simple principle does not work without properly chosen ansätze when dealing with a huge parameter space [5]. For this reason, researchers often incorporate an *inductive bias* into their algorithms [6]. An inductive bias is a prior knowledge about the system under investigation that can be included in the algorithm to restrict our function classes. Thus, the parameterised function favours a better class of outputs for a given target problem. In classical machine learning, for example, it is known that the great success of convolutional neural networks (CNNs) is based on the fact that they contain 'layers', essentially parameterised maps, which encode the idea that the content of an image does not change when shifted. Specifically, these convolutional layers are (approximately) translation equivariant: When one shifts the input state by $n$ pixels up and $m$ bits down, the output is also shifted in the same way [7,8]. Geometric deep learning naturally extends this framework to arbitrary groups [9], suggesting the use of group equivariant layers for learning data with symmetric properties. Neural networks consisting of group equivariant layers have indeed reported better performance for classifying images [7], point clouds [10], and in the modelling

of dynamical systems [11]. More broadly, they have also been used in a general variational context for tasks such as identifying the ground state of molecules [12].

Recently, the idea of geometric machine learning has been combined with quantum machine learning (QML). Generally speaking, QML algorithms [13] hope to find an advantage over classical algorithms in ML tasks by exploiting the quantum nature of Hilbert space using parameterised quantum circuits. Despite its potential, however, the trainability and generalisation performance of QML algorithms without tailored circuit ansätze often scale poorly, limiting their usability for more than tens of qubits [14]. Because of this, recent studies introduced geometric quantum machine learning (GQML) as a guiding principle for constructing a quantum circuit ansatz. The literature shows these symmetry-informed circuits have been successful in offering better trainability and generalisation performance [15–26].

In the GQML setup, the symmetry group SU(2) is particularly interesting as it naturally arises in quantum systems with rotational symmetry. It also corresponds to a natural symmetry of qubits, which can be seen as a product of spin-$\frac{1}{2}$ states. While QML algorithms with the SU(2) symmetry have been previously studied in Refs. [22,24,26], implementing the proposed circuit ansätze in quantum hardware was not straightforward. For example, Ref. [24] proposed twirling as a constructive principle for equivariant gates, but computing this twirling formula for a many-qubit gate is highly non-trivial as it involves the summation over the symmetric group (thus over $n!$ terms). In contrast, Ref. [26] showed that a certain form of elements in an algebra generated by the symmetric group (formally written as $\mathbb{C}[S_n]$) can be seen as SU(2) equivariant quantum circuits. Nonetheless, these circuits do not admit a simple decomposition to few-qubit gates (implementable on quantum hardware).

In this paper, we propose an alternative approach to construct SU(2) equivariant circuits. Our circuit ansätze, dubbed *spin-network circuits*, are inspired by spin networks, SU(2) equivariant tensor networks. A core tool for us will be the *Schur* gate (or map; we will use these terms interchangeably) that sends us from a qubit basis to a spin-basis. For example for two qubits, it provides the following mapping $|J = 0, J_z = 0\rangle = |01\rangle - |10\rangle$, $|J = 1, J_z = 1\rangle = |00\rangle$, $|J = 1, J_z = 0\rangle = |01\rangle + |10\rangle$, and $|J = 1, J_z = -1\rangle = |11\rangle$ where $J$ is the total angular momentum of two qubits and the $J_z$ is its $z$-direction component. The advantage of this basis is that it leaves the matrix representations block-diagonal in the total angular momenta [27]. We use this by applying certain unitaries to these blocks that allow us to directly parameterise the equivariant maps that make up spin networks. This approach to parameterising equivariant maps via their block decomposition as a QML method coincides directly with what is highlighted in Refs. [22,28].

Furthermore, we prove that our circuit is mathematically equivalent to other constructions using the representation theory of SU(2). In particular, we prove that both our gates and gates from the twirling formula [22,24] can be written in the form of generalised permutations as introduced in Refs. [20,26]. When restricted to unitary operators, all three constructions give the same set of gates. Our main theoretical tool is the Schur-Weyl duality, which, roughly speaking, posits a duality between SU(2) and the symmetric group $S_n$. While Refs. [19,22,28] already introduced a general theory of equivariant circuits for arbitrary Lie groups, thus presenting a part of our results in a slightly different manner, we develop a theory specifically for the SU(2) group and provide a concrete example using the three-qubit equivariant gate.

We additionally show that the proposed three-qubit gates can be useful for solving a real-world problem with supporting numerical results for SU(2) symmetric models. While our circuits can be used for usual machine learning tasks, e.g., classifying rotationally invariant data, we choose the problem of finding the ground state of SU(2) symmetric Hamiltonians as it pro-

vides a better benchmark platform for classically simulated QML models (with $\sim 20$ qubits). In particular, we solve the Heisenberg model on one-dimensional triangular and Kagome lattices, which have the SU(2) symmetry but are tricky for Monte Carlo-based classical algorithms due to the sign problem [29, 30]. We show that our circuit ansätze give accurate ground states with a common parameter optimisation technique, which demonstrates the efficiency of our method and justifies the use of our SU(2) equivariant circuits for appropriately symmetric variational and QML problems more generally.

The paper is organised as follows. In Sec. 2, we introduce the preliminaries needed to understand the other sections: The representation theory for SU(2), spin coupling, and spin networks. In Sec. 3, we introduce our ansätze termed *spin-network circuits*, which are param-eterisable unitary quantum circuits that are also spin networks. To this end, the Schur gate will be introduced, a core technical component in creating our parameterisations. We also concretely present the two and three-qubit unitary *vertex* gates. In Sec. 4, we show that all SU(2) equivariant unitaries are a form of generalised permutation. This directly connects the work here with that on permutational quantum computing (PQC) [31, 32] and in particular PQC+ as outlined in Ref. [26]. We also discuss the relation with the twirling method intro-duced in Ref. [24] showing how all SU(2) equivariant gates, i.e., generalised permutations, are the same as the set of all unitary gates generated by twirled Hermitian operators. Next, in Sec. 5, we present the efficacy of the introduced vertex gates by solving the Heisenberg model defined on the one-dimensional triangular lattice and the two-dimensional Kagome lattice. We then discuss the implications of our results and the connections to the broader literature with a particular focus on PQC+ and loop quantum gravity in Sec. 6 and conclude with a short remark in Sec. 7.

Overall, the new contributions of this work are the following: We introduce an SU(2) equiv-ariant quantum circuit ansatz based on spin networks. We provide a number of numerical sim-ulations validating their efficacy, particularly by solving the Heisenberg model on the Kagome lattice. We connect the theory of equivariant operators as seen in the geometric quantum machine learning literature [22] to the work done on PQC+ [20].

## 2 Preliminaries

**Groups and their representation**   Throughout the paper, we are interested in equivariant quantum gates under the SU(2) group transformation. The group SU(2) itself is part of a larger class of groups known as SU($N$) and is a set of $N \times N$ unitary matrices with a determinant of 1. Formally, we can define an SU(2) equivariant gate as a quantum gate $T$ satisfying

$$U^{\otimes n} T = T U^{\otimes n}, \tag{1}$$

for all $U \in \mathrm{SU}(2)$, where $n$ is the number of qubits in a circuit.

If we consider a circuit $C$ constructed with those gates, thus satisfying $C U^{\otimes n} = U^{\otimes n} C$, one can create an SU(2)-invariant output state given an SU(2)-invariant input state. If $|\psi_0\rangle$ is an input state satisfying $|\psi_0\rangle = U^{\otimes n}|\psi_0\rangle$ (we will see an example of such states in Sec. 3), we have

$$U^{\otimes n} C \, |\psi_0\rangle = C U^{\otimes n} \, |\psi_0\rangle = C \, |\psi_0\rangle . \tag{2}$$

Thus, such a circuit $C$ can be used for learning tasks involving rotationally invariant data, e.g., finding ground states of Heisenberg spin models or classifying point sets.

134    The symmetry we consider here is tightly connected to groups and their representation.
135 Recall that a group $G = \{g_i\}$ is a set with a map acting on two of its elements $g_1 \cdot g_2 = g_3$ such
136 that there is an identity $e \cdot g = g$, the operations are associative $g_1 \cdot (g_2 \cdot g_3) = (g_1 \cdot g_2) \cdot g_3$,
137 and there is an inverse for all elements $g \cdot g^{-1} = e$. It is also natural to consider the action of a
138 group on a vector. For example, a rotation $R \in \mathrm{SO}(3)$ acts on a three-dimensional (real) vector
139 and transforms it. This type of action (on a vector space) is called a *representation* of a group.

140    Formally speaking, a group representation is a map $R : G \rightarrow \mathrm{GL}(V)$ from the group to the
141 space of invertible linear maps of a vector space $V$ (or equivalently, invertible matrices of
142 dimension $N$ if $\dim(V) = N$) such that $R(g_1 \cdot g_2) = R(g_1) \cdot R(g_2)$. In essence, it is a map from
143 the group to linear maps that preserves the group structure. For a system with a single qubit, a
144 simple map $R(U) = U$ for $U \in \mathrm{SU}(2)$ already defines a representation. One can readily extend
145 this representation to a $n$-qubit system by defining $\widetilde{R}(U) = U^{\otimes n}$, which is also a representation
146 (as $\widetilde{R}(U_1 U_2) = (U_1 U_2)^{\otimes n} = U_1^{\otimes n} U_2^{\otimes n} = \widetilde{R}(U_1)\widetilde{R}(U_2)$). We can then see that to find $\mathrm{SU}(2)$
147 equivariant gates for an $n$-qubit system, we must pay attention to the representation $\widetilde{R}$.

148    Studying the representation of symmetry introduces the concept of *irreducible representa-*
149 *tions* (irreps, for short). Firstly, a sub-representation $W$ of $V$ is a subspace $W \leq V$ which
150 satisfies $R(g)W = \{R(g)w : w \in W\} \subseteq W$ for all $g \in G$. Then we say a representation
151 $R : G \rightarrow \mathrm{GL}(V)$ is irreducible if it does not have any non-trivial sub-representations, i.e. if
152 $W \leq V$ and $R(g)W = \{R(g)w : w \in W\} \subseteq W$ for all $g \in G$, then $W = 0$ or $W = V$. Thus, we
153 may find a structure of equivariant gates by decomposing the $n$-qubit system to vector spaces
154 of different spin numbers (which is always possible by the Peter–Weyl theorem). As we shall
155 see, the *Schur map* sends equivariant operators into a block diagonal form. This form will
156 allow us to design such maps explicitly.

157 **From qubits to spins**    A spin is an irreducible representation of the $\mathrm{SU}(2)$ group. This vector
158 space is spanned by basis vectors $\{|J, J_z\rangle : -J \leq J_z \leq J\}$ where $2J$ is an integer (e.g., $J = 0$,
159 $J = \frac{1}{2}$, $J = 1$, $J = 3/2$, etc.). Physically, $J$ and $J_z$ correspond to the quantised total angular
160 momentum and the angular momentum in the $z$-direction, respectively (though the $z$-direction
161 is a conventional choice, any would do). For each allowed value of $J$, we call the corresponding
162 vector space a spin-$J$ system.

163    A qubit is naturally identified as a spin-$\frac{1}{2}$ particle, by a mapping $|0\rangle = |J = \frac{1}{2}; J_z = \frac{1}{2}\rangle$ and
164 $|1\rangle = |J = \frac{1}{2}; J_z = -\frac{1}{2}\rangle$. When we take two qubits, we are thinking of the basis elements
165 $\{|00\rangle, |01\rangle, |10\rangle, |11\rangle\}$. Consider the angular momentum of two qubits (or two spin-$\frac{1}{2}$ par-
166 ticles, equivalently). It is well known that when one considers two spin-systems of momenta
167 $J_1$ and $J_2$ in terms of their joint angular momentum, the possible total angular momentum $J$
168 measurements range from $J = |J_1 - J_2|$ to $J_1 + J_2$. Thus, two qubits have the two total angular
169 momentum possibilities of $J = 0$ and $J = 1$. To get the full basis, we must include the possible
170 $J_z$ values ranging from $-J$ to $J$ in steps of 1 [33]. In general, we can always move from a basis
171 of qubits to a basis of angular momenta by considering the pairwise coupling of qubits and
172 subsequent spins, which amounts to considering the possible angular momentum outcomes of
173 a measurement of each pairing. This coupling scheme is depicted in Fig. 1.

174    For more than two spins, we will have a choice of the order in which we do this. The different
175 orders of pairing the spin systems amount to different bases (as they correspond to different
176 choices of complete measurements), which we can describe by branching tree-like structures.
177 In Fig. 2, we can see this for three qubits.

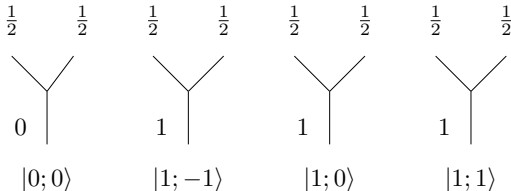

Figure 1: Graphical presentation of the basis constructed by combining angular momentum of two spin-$\frac{1}{2}$ systems and the possible outcomes of total and $z$-directed angular momenta. These can be seen as two spin networks, corresponding to the two possible total angular momentum values on the bottom edge, with specific $|J; J_z\rangle$ states chosen for the bottom edges Hilbert spaces.

In later discussion, we will use $J_{\mathcal{J}} = \mathbb{C}^{2\mathcal{J}+1}$ to denote a spin-$\mathcal{J}$ system. For example, $J_{1/2} = \mathbb{C}^2$ is a vector space for spin-$\frac{1}{2}$ system, i.e., a qubit.

**Spin networks**   We now consider a generalisation of equivariant gates using multi-linear maps. Let us first recall properties of spin-1/2 kets and bras under $g \in \mathrm{SU}(2)$:

$$|a\rangle \xrightarrow{g} g|a\rangle \tag{3}$$

$$\langle b| \xrightarrow{g} \langle b|g^\dagger, \tag{4}$$

where $g = e^{-i\phi \boldsymbol{\sigma} \cdot \hat{n}/2} \in \mathrm{SU}(2)$. Here, $\boldsymbol{\sigma} = \{\sigma_x, \sigma_y, \sigma_z\}$ is a vector of $2 \times 2$ Pauli matrices, $\hat{n}$ is a normal vector indicating the direction of the rotation, and $\phi$ is the angle we rotate.

By identifying kets as vectors and bras as dual vectors, we can generalise the above principle by considering an arbitrary spin-$\mathcal{J}$ system given as $V = J_{\mathcal{J}} = \mathbb{C}^{2\mathcal{J}+1}$. Then $|a\rangle \in V$ and $\langle b| \in V^*$ changes to

$$|a\rangle \xrightarrow{g} R(g)|a\rangle \tag{5}$$

$$\langle b| \xrightarrow{g} \langle b|R(g)^\dagger \tag{6}$$

under the group transformation, where $R(g)$ is a representation of $g \in \mathrm{SU}(2)$. Specifically, it is a $2\mathcal{J}+1$ by $2\mathcal{J}+1$ unitary matrix given by $e^{-i\phi \boldsymbol{J} \cdot \hat{n}}$ which is a representation of $e^{-i\phi \boldsymbol{\sigma} \cdot \hat{n}/2} = g \in \mathrm{SU}(2)$. Here, $\boldsymbol{J} = \{J_x, J_y, J_z\}$ is a vector of $2\mathcal{J}+1$ by $2\mathcal{J}+1$ spin matrices satisfying $[J_a, J_b] = i\epsilon_{abc}J_c$ for all $a, b, c \in \{x, y, z\}$ where $\epsilon_{abc}$ is the Levi-Civita symbol.

The above principle also induces group transformation formulas for other expressions. For example, one can see that the inner product $\langle a|b\rangle$ is invariant under the group transform as

$$\langle b|a\rangle \xrightarrow{g} \langle b|R(g)^\dagger R(g)|a\rangle = \langle b|a\rangle. \tag{7}$$

Note that the last equality is obtained as $R(g)$ is unitary. Next, let us consider a linear map $T : V \to V$. As $T$ can be written as $T = \sum_{ij} t_{ij} |i\rangle \langle j| \in V \otimes V^*$, we know it changes to

$$T \xrightarrow{g} R(g)TR(g)^\dagger \tag{8}$$

under the transformation.

We now add a constraint that a linear map $T$ also preserves the group structure. In other words, we require $T$ to satisfy

$$R(g)(T|a\rangle) = T(R(g)|a\rangle) \tag{9}$$

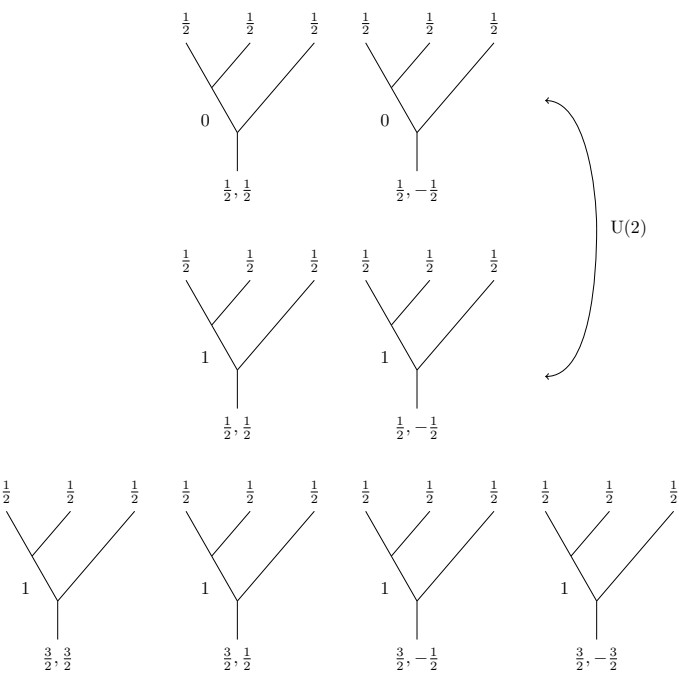

Figure 2: Graphical depiction of a coupling basis of three qubits, where the pairwise coupling of the spaces proceeds from the left (other possibilities give alternative bases). Each row of trees is indexed by the possible total angular momenta that can occur for each composition of two systems. The elements in the rows correspond to the different states, giving a final $J_z$ value on the spaces at the bottom of the trees. Note how the top two rows of diagrams index spaces with the same total angular momentum at the base but that the patterns of coupling that form them are distinct. In Sec. 4, we will see that this allows for the mixing of such states because SU(2) equivariant maps cannot distinguish the two spin coupling structures. Note that in the absence of specifying the $J_z$ values, the set of diagrams on each row correspond to three separate spin networks as the SU(2) invariance on three-valent networks reduces to spin-coupling rules; this is discussed in more detail in Appendix A.

198  for all $g \in G$ and $|a\rangle \in V$, which implies that $R(g)^\dagger T R(g) = T$ (or equivalently, $T = R(g)TR(g)^\dagger$).
199  As $R(g)TR(g)^\dagger$ is nothing but $T$ after the group transformation, a linear map preserving the
200  group structure is a matrix that is invariant under the group transformation (given by conju-
201  gation with $R(g)$).

202      One may further extend this property to multilinear maps (tensors). For example, a two-
203  qubit gate is a linear map $T$ between $V^{\otimes 2}$ and $V^{\otimes 2}$ (where $V = J_{1/2} = \mathbb{C}^2$ in the standard
204  formulation). If we add the equivariant condition to this gate, i.e., $R(g)^{\otimes 2} T = T R(g)^{\otimes 2}$, this
205  is nothing but the condition for a group-structure preserving map. As a two-qubit gate $T$ can
206  be considered as an element of $V^{\otimes 2} \otimes (V^*)^{\otimes 2}$, $T$ becomes

$$T \xrightarrow{g} R(g)^{\otimes 2} T (R(g)^\dagger)^{\otimes 2} = T, \tag{10}$$

207  under the group transformation, where the last equality is from the equivariant condition.
208  Thus there is one-to-one correspondence between group-structure preserving maps and group-
209  invariant tensors[1]. In other words, if we consider a general (possibly non-unitary) linear map
210  between $V^{\otimes n}$ and $V^{\otimes m}$ (where $n$ and $m$ can be different integers), preserving the group struc-

---

[1]Formally, the set of these tensors is written as $\mathrm{Inv}_{\mathrm{SU(2)}}(V^{\otimes n} \otimes (V^*)^{\otimes m})$.

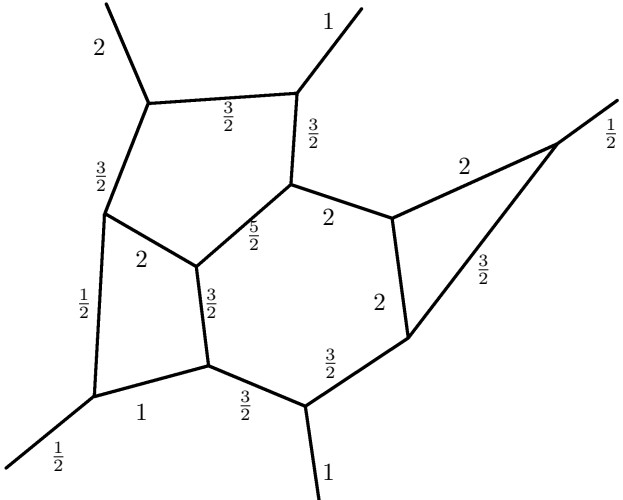

Figure 3: A three-valent spin network typically presented in the broader literature: an edge-labelled graph (though directed, this is often suppressed in depictions since the spaces are isomorphic). In the three-valent case, the edge labels are spins such that around any vertex they meet the Clebsch-Gordan conditions $j_1 + j_2 + j_3 \in \mathbb{N}$ and $|j_1 - j_2| \leqslant j_3 \leqslant j_1 + j_2$. which can be shown to exactly match when the vertex is an invariant subspace of SU(2) (See Appendix A for more details).

ture, it can be seen as a group-invariant tensor with $n$ input legs and $m$ output legs [34, 35] (often called a tensor of type $(n, m)$).

Now, we consider a tensor network that consists of SU(2) invariant tensors with contraction edges that run over irreps of SU(2). This special type of network is called a "spin network"; an example from the broader literature can be seen in Fig. 3. These were originally introduced by Penrose [36] in the very different context of a combinatorial derivation of space-time. In modern physics, they are typically discussed as the basis of quantised space in the covariant formulation of loop quantum gravity [37] (though not the focus of this work, interested readers can look Appendix C for the connection). Roughly, a spin network is a directed graph where each edge has an associated spin, and each vertex $v$ has an associated equivariant map from the tensor product of the incoming spins to the tensor product of the outgoing spins. Formally, we describe this as a graph detailing the connectivity of vertices $v$ with incoming edges $e_{in}$ and outgoing ones $e_{out}$ such that for every vertex, there is an associated map $T_v$ such that $T_v \in \bigotimes_{i \in e_{in}} \bigotimes_{o \in e_{out}} J_{j_i} \otimes J_{j_o}^*$, where $J_{j_i}$ and $J_{j_o}$ are the incoming and outgoing respective Hilbert spaces. We further require $T_v$ to satisfy the equivariant condition

$$\bigotimes_{i \in e_{in}} \bigotimes_{o \in e_{out}} T_v \left( R_{j_i}(g) J_{j_i} \otimes J_{j_o} \right) = \bigotimes_{i \in e_{in}} \bigotimes_{o \in e_{out}} T_v \left( J_{j_i} \otimes R_{j_o}(g) J_{j_o} \right) \qquad \forall g \in G, \quad \forall v, \qquad (11)$$

where $R_{j_i}(g)$ and $R_{j_o}(g)$ are the representations of the group element $g$ acting on the $J_{j_i}$ and $J_{j_o}$, respectively. From the discussion above, each map associated with a vertex ($T_v$) can be regarded as a group-invariant tensor. In this way, spin networks are tensor networks where the composing tensors are elements in the invariant sub-spaces of a group, and the contraction is over spin spaces of size $2J + 1$. For a more detailed description of these objects, we direct the reader to Appendix A. For our interests, it is sufficient to say that we can build a quantum circuit that is inherently SU(2) equivariant by restricting to specific spin networks whose vertices can

be interpreted as parameterised qubit unitaries.

Within the literature, spin networks that form binary trees have been particularly promi-
nent. The simplest example is seen in Fig. 1, where we ignore the specification of the $J_z$ state
at the bottom and focus only on the total angular momentum (so there are just two unique
diagrams from this perspective). A more general example is provided by Fig. 2, where we
have three spin spaces coming together, which naturally leads to three possible spin networks,
specifically one for each row. The columns are not different networks because they amount
to fixing a choice of $J_z$ value on one edge, which is a choice of contraction index (i.e., final
projection). Thus, such a fixing does not alter the spin-spaces in the definition of the network[2].
It should be noted that spin networks have previously been considered in the broader quan-
tum information literature as diagrammatic qubit maps and as variational maps for numerical
investigations of LQG on quantum computers Refs. [38–41] though never as general SU(2)
equivariant variational ansätze.

# 3   Spin-network circuits

In this section, we outline circuit ansätze designed based on the principles of spin networks.
To show their utility, we present concrete examples, which in turn are used for our simulations
further below in Sec. 5. Due to the circuits' mathematical equivalence to certain types of spin
networks, they are explicitly SU(2) equivariant. While the core ideas are outlined here, we
discuss the finer points, related concepts, and generalisations in Appendix A.

Our circuits, termed *spin-network circuits*, are a specific form of spin network. They are
spin networks where all vertices have an even number of external wires, and every wire in
the network is spin-$\frac{1}{2}$, and so are formed of qubits. Among all external wires for each vertex,
half are inputs, and the other half are outputs; the combination of these vertices amounts to a
quantum circuit. For this reason, when viewed as a quantum circuit, we refer to the vertices
as *vertex gates*. Critically, the vertices of a spin network are equivariant maps between the
input and output edges, which is a direct consequence of the definition given in Eq. (11). This
means the resultant circuit is also equivariant. An important property of spin networks with
vertices with more than three edges is that they can be parameterised (see Appendix A). By
training over these parameters, we thus arrive at a trainable equivariant network.

**Schur gate and two-qubit vertex gate**   The simplest spin-network circuit is built from ver-
tex gates acting solely on two qubits. To understand the structure of this gate and its later
generalisations, we first require the two-qubit Schur gate as a prerequisite [42]:

$$S_2 = \begin{pmatrix} 1 & 0 & 0 & 0 \\ 0 & \frac{1}{\sqrt{2}} & \frac{1}{\sqrt{2}} & 0 \\ 0 & 0 & 0 & 1 \\ 0 & \frac{1}{\sqrt{2}} & -\frac{1}{\sqrt{2}} & 0 \end{pmatrix} \tag{12}$$

---

[2]The careful reader might note that here we are simultaneously looking at diagrams that correspond to the
rules of angular momentum addition and saying these match to the definition of the vertices being SU(2) invariant
sub-spaces. The connection is outlined in Appendix C, where we see that the invariant spaces can be decomposed
in terms of Clebsch-Gordan coefficients, which are the exact same elements used in deriving angular momentum
decompositions.

265      This gate is a unitary operator that maps the computational basis of two qubits to the spin
266   basis of their combined $J$ and $J_z$ angular momenta. As qubits can be seen as spin-$\frac{1}{2}$ spaces, with
267   spin-up and spin-down being assigned to 0 and 1 respectively, then qubit registers correspond
268   to tensor products of spin-$\frac{1}{2}$ irreps. While these are individually irreducible, their product is not
269   and can be block-diagonalised into irreducible components. In the case of two qubits, it is often
270   typical to write that $J_{\frac{1}{2}} \otimes J_{\frac{1}{2}} \simeq J_0 \oplus J_1$ which says that a tensor product of two spin-$\frac{1}{2}$ spaces is
271   isomorphic to the direct sum of a spin-0 and a spin-1 space telling us that there is a unitary map
272   between them. The two-qubit Schur gate performs exactly this map. Looking at this in terms of
273   the computational basis, the two-qubit Schur gate maps the computational basis states to the
274   following basis (where we often drop the normalisation in later exposition): $|J = 1, J_z = 1\rangle =$
275   $|00\rangle$, $|J = 1, J_z = 0\rangle = \frac{1}{\sqrt{2}}(|01\rangle + |10\rangle)$, $|J = 1, J_z = -1\rangle = |11\rangle$, and $|J = 0, J_z = 0\rangle = \frac{1}{\sqrt{2}}(|01\rangle -$
276   $|10\rangle)$, which is occasionally referred to as the triplet/singlet basis[3]. In general, though trivially
277   in the two-qubit case, we can say that the two-qubit Schur map sends us to the sequentially
278   coupled basis of two qubits exactly as depicted in Fig. 1. As was discussed in Sec. 2 above, this
279   amounts to two different binary spin networks with the $J_z$ values specified on the base as first
280   outlined in Ref. [43].

281      The two-qubit Schur gate from Eq. (12) is the simplest Schur map that sends us from the
282   tensor product of qubits to the direct sum of spins. Precisely, the general form of the Schur
283   map follows the prescription:

$$S_n : J_{\frac{1}{2}}^{\otimes n} \to \bigoplus_k J_k \tag{13}$$

284   where we understand $J_{\frac{1}{2}}^{\otimes n}$ as the Hilbert space corresponding to $n$ qubits and $k$ ranges over the
285   irreducible representations of SU(2) that make up the space in the spin-basis where we note
286   that *irreps can repeat, in which case we say there is a multiplicity*[4].

287      The matrix elements of the Schur map can be obtained by using *Clebsch-Gordan coefficients*
288   and coupling paths of qubits. Each Clebsch-Gordan coefficient $\langle j_1 m_1 j_2 m_2 \,|\, JM \rangle = c^{JM}_{j_1 m_1 j_2 m_2}$
289   corresponds to the projection of two particular spin-states into their combined angular mo-
290   menta. Thus, its matrix entries correspond to the Clebsch-Gordan coefficients that result from
291   projecting coupled spin systems (specifically one spin-$\frac{1}{2}$ with whatever angular momentum
292   that previous spin-couplings have reached) into a particular total $J$ value. Each coefficient
293   that gets multiplied corresponds to a vertex in the coupling diagrams that index each of the
294   spin-basis elements (such as those seen in Fig. 2), i.e., each element of the Schur map can
295   be obtained by multiplying the Clebsch-Gordan coefficients associated with each vertex of the
296   spin-coupling diagram.

297      As an example, let us consider the three-qubit case. Here each element in the matrix of the
298   Schur map corresponds to $c^{j', m'}_{j_1, m_1; j_2, m_2} c^{J, M}_{j', m'; j_3, m_3}$ for some choice of $j' \in \{0, 1\}$ and $-j' \le m' \le j'$.
299   Here $j'$ stands for the resulting spin from coupling the first two qubits, which leads to possible
300   total spin momenta $j' = 0$ and $j' = 1$. In the following, we focus on the spin-0 case ($j' = 0$).
301   This corresponds to the coefficient $c^{0,0}_{\frac{1}{2}, m_1; \frac{1}{2}, m_2}$. When we, in turn, couple with the third qubit
302   the only possible outcome for the total angular momentum is $\frac{1}{2}$, so the combined coupling
303   coefficient for these total angular momenta is $c^{0,0}_{\frac{1}{2}, m_1; \frac{1}{2}, m_2} c^{\frac{1}{2}, m}_{\frac{1}{2}, m_1; 0, 0}$. These choices single out a
304   particular recoupling path with associated final $J_z$ values on the root (as seen in Fig. 1) and
305   so a row in the matrix. The computational basis, equivalently the $J_z$ values for the individual

---

[3]For reasons of the different total angular momentum states energies separating under the presence of an external magnetic field.

[4]More formally the Schur map implements the isomorphism given in Theorem 2 below.

306 qubits, fixes the columns (for more on this, see Ref. [44]). For practical implementations,
307 it is important to note that the Schur gate can be implemented in polynomial time, and the
308 literature already contains examples of specific methods to do this [44, 45].

309 In the case of two qubits, there is only a single coefficient to consider in each element of the
310 matrix, and so we have the following:

$$
S_2 = \begin{pmatrix}
c^{1,1}_{\frac{1}{2},\frac{1}{2},\frac{1}{2},\frac{1}{2}} & c^{1,1}_{\frac{1}{2},\frac{1}{2},\frac{1}{2},-\frac{1}{2}} & c^{1,1}_{\frac{1}{2},-\frac{1}{2},\frac{1}{2},\frac{1}{2}} & c^{1,1}_{\frac{1}{2},-\frac{1}{2},\frac{1}{2},-\frac{1}{2}} \\
c^{1,0}_{\frac{1}{2},\frac{1}{2},\frac{1}{2},\frac{1}{2}} & c^{1,0}_{\frac{1}{2},\frac{1}{2},\frac{1}{2},-\frac{1}{2}} & c^{1,0}_{\frac{1}{2},-\frac{1}{2},\frac{1}{2},\frac{1}{2}} & c^{1,0}_{\frac{1}{2},-\frac{1}{2},\frac{1}{2},-\frac{1}{2}} \\
c^{1,-1}_{\frac{1}{2},\frac{1}{2},\frac{1}{2},\frac{1}{2}} & c^{1,-1}_{\frac{1}{2},\frac{1}{2},\frac{1}{2},-\frac{1}{2}} & c^{1,-1}_{\frac{1}{2},-\frac{1}{2},\frac{1}{2},\frac{1}{2}} & c^{1,-1}_{\frac{1}{2},-\frac{1}{2},\frac{1}{2},-\frac{1}{2}} \\
c^{0,0}_{\frac{1}{2},\frac{1}{2},\frac{1}{2},\frac{1}{2}} & c^{0,0}_{\frac{1}{2},\frac{1}{2},\frac{1}{2},-\frac{1}{2}} & c^{0,0}_{\frac{1}{2},-\frac{1}{2},\frac{1}{2},\frac{1}{2}} & c^{0,0}_{\frac{1}{2},-\frac{1}{2},\frac{1}{2},-\frac{1}{2}}
\end{pmatrix} = \begin{pmatrix}
1 & 0 & 0 & 0 \\
0 & \frac{1}{\sqrt{2}} & \frac{1}{\sqrt{2}} & 0 \\
0 & 0 & 0 & 1 \\
0 & \frac{1}{\sqrt{2}} & -\frac{1}{\sqrt{2}} & 0
\end{pmatrix}
$$

311 which indeed matches the definition of the two-qubit Schur gate in Eq. (12).

312 Once we are in the spin basis, we can elegantly construct the two-qubit vertex gate by ap-
313 plying a phase solely on the spin-0, or singlet, element $|J = 0, J_z = 0\rangle$ (see Lemma 1 below).
314 Intuitively, suppose a map is SU(2) equivariant so that you can isolate and apply group repre-
315 sentations before or after the map. In that case, the different spin-irreps should not interact
316 under the mapping and remain differentiated – as matrices. This is why the map is block di-
317 agonal in the spin basis. For the two-qubit case, up to a global phase, this amounts to just a
318 phase on one of the spaces:

$$
P_2(\theta) = \left(\begin{array}{ccc|c}
& & & 0 \\
& \mathbb{1}_3 & & 0 \\
& & & 0 \\
\hline
0 & 0 & 0 & e^{i\theta}
\end{array}\right)
\tag{14}
$$

319 In terms of spin networks, which we recall are equivariant maps, the Schur gate is sending us
320 to the two possible coupling options. Two qubits coupling to spin-0 or to spin-1. In isolation[5],
321 these correspond to two possible spin networks. The parameterised gate $P_2(\theta)$ applies a phase
322 on the spin-0 network. In Sec. 4, this structure completely characterises the possible unitary
323 equivariant maps. To understand how this phase manages to isolate only one part of the spin
324 space, we need to look again at representations. The spin basis is always such that any group
325 representation in this basis (up to row permutation depending on your exact basis choices
326 and Schur gate, which can vary a little in the literature) is block diagonal. Each individual
327 block is associated with a particular total angular momentum $J$ *and* a way of arriving at it
328 by sequentially coupling spin-1/2s as seen in Fig. 2. In this way, given a tensor product of $n$-
329 spins, each block corresponds to one of the $2J + 1$ dimensional spin spaces of its direct product
330 decomposition as seen in Eq.(13). As we now know, for the case of two qubits, we either have
331 spin-0 or spin-1, and so this block decomposition resembles the following:

$$
\left(\begin{array}{ccc|c}
& & & 0 \\
& \text{spin-1} & & 0 \\
& & & 0 \\
\hline
0 & 0 & 0 & \text{spin-0}
\end{array}\right)
\tag{15}
$$

332 The block diagonal structure is critical for our SU(2) equivariant ansätze. As we will see
333 below, their general structure is to apply parameterised maps that act independently on blocks
334 of different sizes (which are different irreducible representations) and as unitaries that mix

---

[5]An equivariant gate acting on two or more qubits can be regarded as a spin network with more than three legs.
One can specify intermediate vertex choices for such a network, which introduces a sub-network structure.

$$V(\theta) \in \text{Inv}_{\text{SU(2)}}(J_{\frac{1}{2}} \otimes J_{\frac{1}{2}} \otimes J_{\frac{1}{2}} \otimes J_{\frac{1}{2}}) \quad = \quad |J = 1\rangle\langle J = 1| \oplus e^{i\theta}|J = 0\rangle\langle J = 0|$$

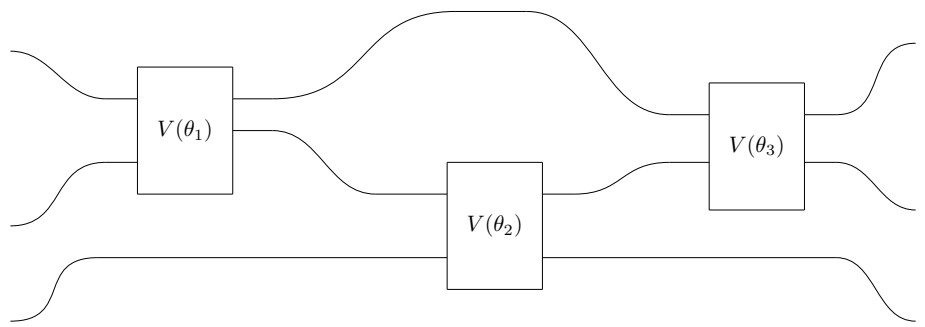

Figure 4: Depiction of a parameterised gate $V(\theta) \in \text{Inv}_{\text{SU(2)}}(J_{\frac{1}{2}} \otimes J_{\frac{1}{2}} \otimes J_{\frac{1}{2}} \otimes J_{\frac{1}{2}})$ living in the basis block diagonal in the space of SU(2) equivariant unitaries on two qubits and therefore a four-valent spin network vertex. It is composed of a superposition of two three-valent spin networks indexed by the possible internal spin-0 or spin-1 edge (see Appendix C for details on spin network decompositions). On the right-hand side, we allude to the geometric interpretation of the basis where the couplings correspond to triangles of different quantised edge length (again see Appendix C).

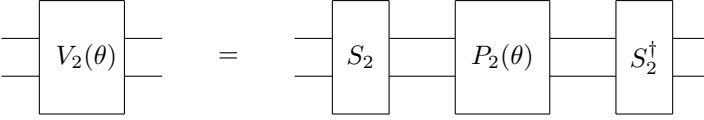

Figure 5: A four-valent spin-network circuit that can be trained over the free parameters in its vertex gates. The curved qubit wires highlight that such spin-network circuits are both spin networks and quantum circuits.

those parts of repeated blocks of the same irreducible representation when they correspond to the same $J_z$ value. Indeed, this structure completely characterises equivariant maps, as is shown below in Sec. 4. As such, we can create an equivariant ansatz for SU(2), i.e., spin rotation symmetry. We note it resembles work seen in Ref. [22].

This leads us to the definition of a vertex gate.

**Definition 1.** *The two-qubit vertex gate $V_2(\theta)$ is composed as follows:*

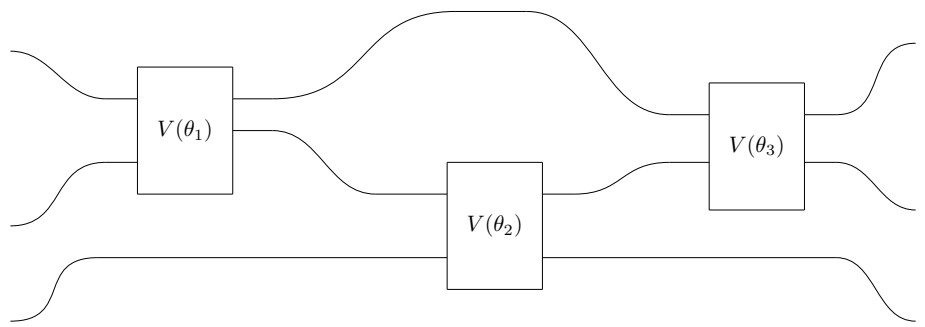

*where $S_2$ is the two qubit Schur gate and $P_2(\theta)$ is the controlled phase seen in Eq. (14).*

What we have created is specific two-qubit gates that live in the space of equivariant maps from, and to, the tensor product of two spin-$\frac{1}{2}$s; these can be seen depicted in Fig. 4. These, by definition, are elements of the vertices of a four-valent spin network with edges fixed as qubits. We can see the spin network as corresponding to an operator formed by sequential gate operations as seen in Fig. 5

**Three and more qubit vertex gates** Every even valence spin network vertex admits a possible vertex gate (though two is trivial; see Appendix C). A second, more subtle, example is the

349    three-qubit Schur gate $S_3$.

$$
S_3 = (c^{j_4,m_4}_{j_1,m_1;j_2,m_2} c^{J,M}_{j_4,m_4;j_3,m_3}) = \begin{pmatrix}
1 & 0 & 0 & 0 & 0 & 0 & 0 & 0 \\
0 & \frac{1}{\sqrt{3}} & \frac{1}{\sqrt{3}} & 0 & \frac{1}{\sqrt{3}} & 0 & 0 & 0 \\
0 & 0 & 0 & \frac{1}{\sqrt{3}} & 0 & \frac{1}{\sqrt{3}} & \frac{1}{\sqrt{3}} & 0 \\
0 & 0 & 0 & 0 & 0 & 0 & 0 & 1 \\
0 & \sqrt{\frac{2}{3}} & -\frac{1}{\sqrt{6}} & 0 & -\frac{1}{\sqrt{6}} & 0 & 0 & 0 \\
0 & 0 & 0 & \frac{1}{\sqrt{6}} & 0 & \frac{1}{\sqrt{6}} & -\sqrt{\frac{2}{3}} & 0 \\
0 & 0 & -\frac{1}{\sqrt{2}} & 0 & \frac{1}{\sqrt{2}} & 0 & 0 & 0 \\
0 & 0 & 0 & -\frac{1}{\sqrt{2}} & 0 & \frac{1}{\sqrt{2}} & 0 & 0
\end{pmatrix}
\tag{16}
$$

350    Again we have a parameterised $P_3(\vec{\theta})$ rotation applied in the spin basis. In the parameterised
351    gate we define a three-qubit unitary that acts on the two spin-$\frac{1}{2}$ spaces that come from the block
352    diagonal decomposition of three qubits $J_{\frac{1}{2}} \otimes J_{\frac{1}{2}} \otimes J_{\frac{1}{2}} \simeq J_{\frac{3}{2}} \oplus J_{\frac{1}{2}} \oplus J_{\frac{1}{2}}$. The difference between this
353    gate and the one above is that the above two-qubit vertex gate lacks multiplicities, i.e., multiple
354    blocks of the same size, meaning the only option is to have a phase on each different block. If
355    we have multiple blocks of the same size, this indicates that there are multiple sub-spaces of
356    the state space with the same total angular momentum and that multiple states exist with the
357    same quantum numbers $|J;J_z\rangle$. In terms of SU(2) equivariant maps, these are states that we
358    can interchange without altering the structure of the space – this implies that our vertex gates
359    are not just phases on differing blocks but also unitaries that mix the multiple copies of $|J;J_z\rangle$
360    (see Fig. 2 for how our unitaries act on this space and Sec. 4 for theoretical backgrounds). As
361    an example, for our three-qubit space, we have one spin-$\frac{3}{2}$ space and two spin-$\frac{1}{2}$ spaces so it
362    suffices to have a single unitary acting to mix the two $|\frac{1}{2},J_z\rangle$ states. The general matrix has
363    the following form:

$$
P_3(\vec{\theta}) = \left(\begin{array}{c|c}
\mathbb{1}_4 & 0_4 \\
\hline
0_4 & U_2(\vec{\theta}) \otimes \mathbb{1}_2
\end{array}\right) = \left(\begin{array}{c|c}
\mathbb{1}_2 & 0_2 \\
\hline
0_2 & U_2(\vec{\theta})
\end{array}\right) \otimes \mathbb{1}_2
\tag{17}
$$

364    where $U_2(\vec{\theta})$ is a unitary matrix of dimension two, implying this gate has four real parameters.
365    One might imagine that there could be a relative phase here on the isolated spin-$\frac{3}{2}$ space but
366    (up to a global phase) this is a sub-case of the unitary acting on the two spin-$\frac{1}{2}$ components. We
367    note that this gate can be written as the ControlledUnitary gate between the first and second
368    qubits (and acting trivially on the third qubit), which is generated by $\{|1\rangle\langle 1| \otimes \mathbb{1}_2, |1\rangle\langle 1| \otimes$
369    $X, |1\rangle\langle 1| \otimes Y, |1\rangle\langle 1| \otimes Z\}$.

370    This leads to the three-qubit vertex gate definition.

371    **Definition 2.** *The three-qubit vertex gate is composed as follows:*

372    *where $S_3$ is the three qubit Schur gate and $P_3(\vec{\theta})$ is the controlled unitary seen in Eq. (17).*

Our construction extends to arbitrary $k$-qubit gates. In general, these spin-network circuits have the following shape:

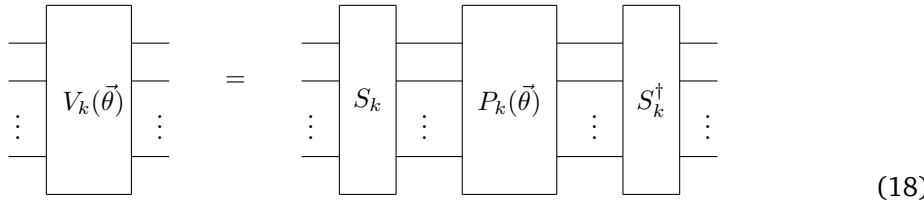

$$(18)$$

Here, $\vec{\theta}$ is the vector of trainable parameters. These are the free variables needed to parameterise the space of the $l$ different irreps that make up the spin basis of $k$ qubits $\oplus_{i=1}^{l}(U_i \otimes \mathbb{1}_{d_i})$ where each $U_i \in U(m_i)$ is unitary of the size of the multiplicity of the $i^{th}$ representation and $d_i$ is the dimension of the $i^{th}$ irrep (i.e., $2J + 1$ where $J$ is the spin number of the subspace). These unitaries mix the states with the same $J_z$ value between the repeated irreps (again see Sec. 4). As any arbitrary $k$-qubit gate can be decomposed into $\mathcal{O}(k)$ elementary gates [46], one can implement a spin-network circuit with a given parameter $\vec{\theta}$ using quantum hardware with a constant overhead (as $k$ is constant). However, it is generally difficult to decompose a spin-network circuit with arbitrary $\vec{\theta}$ to single and two-qubit parameterised quantum gates with a fixed structure, and so this is a compilation task that requires further study (i.e., finding a circuit with single and two-qubit parameterised gates that generate the equivariant gate).

An interesting question is how the few-qubit gates introduced in this section act on the global SU(2) subspace. For example, let us consider a spin-3 irreducible subspace of 8 qubits (e.g., a state $\cos(\theta)|11111110\rangle + \sin(\theta)|11011111\rangle$ lives in this subspace). How can we write down the matrix form of the gate in this subspace? In the following section, we answer this question by outlining the theory of SU(2) equivariant gates from a global perspective. Interestingly, we will show that all SU(2) equivariant gates are the generalised permutations introduced in Ref. [20].

# 4 Equivariant gates from representation theory

In the previous section, we have introduced the Schur map for constructing gates that commute with the SU(2) group action. However, the transformed basis from the Schur map only block diagonalise SU(2) action, and an additional parameterised unitary gate (introduced as $P(\theta)$) acting between the blocks was necessary to build an equivariant gate. In this section, we completely characterise all possible forms of such unitary gates by developing a general theory of SU(2) equivariant operations. Furthermore, using the representation theory of SU(2) and the duality between the permutation group $S_n$ and SU(2), we prove that SU(2) equivariant operations are generalised permutations (which we formally define below), and conversely, all generalised permutations are also equivariant operators. Using this result, we prove that our construction of equivariant gates gives the identical set of gates from the twirling formula and parameterised permutations introduced in Refs. [20, 24]. We further answer the question raised at the end of the previous section using this identification. As this section is rather technical and not directly related to simulation results, the readers may directly jump to later sections.

### 4.1 Equivariant operations as the commutant algebra of a representation

Let us start with the definition of the commutant algebra.

**Definition 3.** *For a given representation $R : T \to \mathrm{GL}(\mathbb{C}^n)$, we define the commutant algebra $C(R)$ as*

$$C(R) = \{T \in \mathcal{M}_n(\mathbb{C}) : TR(g) = R(g)T \text{ for all } g \in G\}, \tag{19}$$

*where $\mathcal{M}_n(\mathbb{C})$ is the set of $n \times n$ complex matrices.*

One can verify that $C(R)$ forms an algebra (under matrix addition and multiplication). This tells us that equivariant gates for $U^{\otimes N}$ with $U \in \mathrm{SU}(2)$ are nothing but unitary operators in $C(U^{\otimes N})$.

Throughout the rest of this subsection, we will construct a complete set of equivariant gates. To achieve this, it will be practical to pay closer attention to the structure of the commutant algebra. To this end, we consider the following lemmas.

**Lemma 1** (Schur's lemma). *A homomorphism preserving the group structure $f \in \mathrm{Hom}_G(V, W)$ is a homomorphism satisfying $f(gv) = gf(v)$ for all $g \in G$ and $v \in V$. If $V$ and $W$ are two irreducible representations of a group $G$ over $\mathbb{C}$, then $f$ must be $c\mathbb{1}$ for $c \in \mathbb{C}$ or $0$.*

In short, a structure-preserving map between two irreps is either proportional to the identity (which implies that the vector space $V$ and $W$ are essentially the same) or zero (they are different irreps). A proof can be found in Refs. [33, 47]. As $T \in \mathrm{Hom}_G(V, W)$ in Definition 3 is a linear map, the condition $TR(g) = R(g)T$ can be written in terms of matrices. From this, we can more easily construct the commutant algebra for some simple cases. For example, the commutant of a direct sum of differing irreps is a direct sum of two scaled identity maps.

**Lemma 2.** *Let $R^{(1)}$ and $R^{(2)}$ be different irreducible representations of a group $G$ with dimensions $d_1$ and $d_2$, respectively. Let us consider a representation $R = R^{(1)} \oplus R^{(2)}$, written as*

$$R(g) = \begin{pmatrix} R^{(1)}(g) & 0 \\ 0 & R^{(2)}(g) \end{pmatrix}. \tag{20}$$

*Then we have*

$$C(R) = \{c_1 \mathbb{1}_{d_1} \oplus c_2 \mathbb{1}_{d_2} : c_1, c_2 \in \mathbb{C}\}. \tag{21}$$

*Proof.* Let $T$ be a matrix with internal blocks $T_{1,1}, T_{1,2}, T_{2,1}, T_{2,2}$ given by

$$T = \begin{pmatrix} T_{1,1} & T_{1,2} \\ T_{2,1} & T_{2,2} \end{pmatrix}. \tag{22}$$

If $TX = XT$,

$$T_{1,1}R^{(1)} = R^{(1)}T_{1,1}, \qquad T_{1,2}R^{(2)} = R^{(1)}T_{1,2},$$
$$T_{2,1}R^{(1)} = R^{(2)}T_{2,1}, \qquad T_{2,2}R^{(2)} = R^{(2)}T_{2,2}.$$

Using Schur's lemma, we obtain $T_{1,1} = c_1\mathbb{1}$, $T_{2,2} = c_2\mathbb{1}$, $T_{1,2} = T_{2,1} = 0$. $\qquad\square$

434 The situation is more complicated in cases where we have a direct sum of the same represen-
435 tation. In this case, the commutant is not simply a direct sum but allows for mixing between
436 the irreps. As we see further below, this will correspond to mixing between elements of the
437 repeated irreps, which are the same.

438 **Lemma 3.** *We now consider a direct sum of the same representation $R = R^{(1)} \oplus R^{(1)}$. Then we*
439 *have*

$$C(R) = \mathcal{M}_2(\mathbb{C}) \otimes \mathbb{1}_{d_1}. \tag{23}$$

440 *Proof.* As before, we write $T \in C(R)$ in a block-diagonal matrix. Then $TR = RT$ gives

$$T_{i,j} R^{(1)} = R^{(1)} T_{i,j}. \tag{24}$$

441 Schur's lemma implies that each $T_{i,j}$ is proportional to $\mathbb{1}$, i.e., $T_{i,j} = c_{i,j} \mathbb{1}$ for $c_{i,j} \in \mathbb{C}$. Thus we
442 have

$$T = \begin{pmatrix} c_{1,1}\mathbb{1} & c_{1,2}\mathbb{1} \\ c_{2,1}\mathbb{1} & c_{2,2}\mathbb{1} \end{pmatrix} = \begin{pmatrix} c_{1,1} & c_{1,2} \\ c_{2,1} & c_{2,2} \end{pmatrix} \otimes \mathbb{1}. \tag{25}$$

443 □

444 Now let us generalise the above results. Let $R$ be a representation of $G$ on $V$. Then Maschke's
445 theorem (for finite groups) or the Peter–Weyl Theorem (for Lie groups) asserts that $V$ is de-
446 composable into a direct sum of irreducible representations

$$V \simeq m_1 R^{(1)} \oplus m_2 R^{(2)} \oplus \cdots m_k R^{(k)}, \tag{26}$$

447 where $mR = R \oplus R \cdots \oplus R$ signifies $m$ repetitions of the same representation, and $\{R^{(i)}\}$ are the
448 different irreducible representations. Applying the above lemmas gives the following theorem.

449 **Theorem 1.** *Under the decomposition given by Eq. (26), the commutant is given by*

$$C(R) = \{\oplus_{i=1}^{k}(M_i \otimes \mathbb{1}_{d_i}) : M_i \in \mathcal{M}_{m_i}(\mathbb{C}) \text{ for all } i\} \tag{27}$$

450 *where each $d_i$ is the dimension of the representation $R^{(i)}$.*

451 Given that a square matrix $M \oplus N$ is unitary iff $M$ and $N$ are both unitary matrices, we obtain
452 the following corollary.

453 **Corollary 1.** *All unitary operators commuting with $R$ are given by*

$$C(R) \cap U(d) = \{\oplus_{i=1}^{k}(U_i \otimes \mathbb{1}_{d_i}) : U_i \in U(m_i) \text{ for all } i\}, \tag{28}$$

454 *where $d = \dim V = \sum_{i=1}^{k} m_i d_i$ is the dimension of $V$.*

455 The Corollary tells us the exact form of intermediate unitary gates $P(\theta)$ we should use for
456 SU(2) equivariant gates, which is evident from the following example.

457 **Example 1.** *For a system with three qubits, we can decompose the space under* SU(2) *as*

$$(\mathbb{C}^2)^{\otimes 3} \simeq J_{3/2} \oplus J_{1/2} \oplus J_{1/2}, \tag{29}$$

458 *where $J_s$ is a space of total spin s with dimension $2s + 1$. Note that the basis transformation from*
459 *the computational basis to the total spin basis is nothing but the Schur transformation given in*
460 *the previous section [Eq. 3]. We can now see that the unitary operators that commute with* SU(2)
461 *are given (up to a global phase) by*

$$\left( \begin{array}{c|c} \mathbb{1}_4 & 0_4 \\ \hline 0_4 & U_2 \otimes \mathbb{1}_2 \end{array} \right), \tag{30}$$

462 *which is the gate we defined in the previous section.*

### 4.2 SU(2) **equivariant gates are generalised permutations**

We now completely characterise SU(2) equivariant gates for $n$ qubits using the above results by computing the multiplicity of each representation. Our main tool is the Schur-Weyl duality, which posits the duality between the irreducible representation of the symmetric group $S_n$ and that of SU(2). Thus, the multiplicity is given by the dimension of the corresponding irreducible representation of $S_n$.

Let us first define two group actions. For $U \in$ SU(2), we define its action on $(\mathbb{C}^2)^{\otimes n}$ as

$$U(|v_1\rangle \otimes |v_2\rangle \otimes \cdots \otimes |v_n\rangle) = |Uv_1\rangle \otimes |Uv_2\rangle \otimes \cdots |Uv_n\rangle, \tag{31}$$

where each $v_i$ is a vector in $\mathbb{C}^2$. In matrix form, this action is nothing but $U^{\otimes N}$.

Another group we consider is the symmetric group $S_n$. For $\alpha \in S_n$, we define

$$\alpha(|v_1\rangle \otimes |v_2\rangle \otimes \cdots \otimes |v_n\rangle) = |v_{\alpha^{-1}(1)}\rangle \otimes |v_{\alpha^{-1}(2)}\rangle \otimes \cdots \otimes |v_{\alpha^{-1}(n)}\rangle. \tag{32}$$

We can also write down a matrix representation of this group action. Let us consider a transposition $\tau = (a, b) \in S_n$ first, which just swaps the $a$-th and $b$-th qubit. In matrix form, this operation is written as

$$\tau = \frac{1}{2}\boldsymbol{\sigma}^a \cdot \boldsymbol{\sigma}^b + \frac{1}{2}\mathbb{1}, \tag{33}$$

where $\boldsymbol{\sigma}^i = \{\sigma_x^i, \sigma_y^i, \sigma_z^i\}$ is a vector of Pauli matrices acting on the $i$-th qubit. As any permutation $\alpha$ in $S_n$ can be decomposed into transpositions, i.e., $\alpha = \tau_k \cdots \tau_2 \tau_1$ where each $\tau_i = (a_i, b_i)$ is a transposition, we obtain

$$\alpha = \left(\frac{1}{2}\boldsymbol{\sigma}^{a_k} \cdot \boldsymbol{\sigma}^{b_k} + \frac{1}{2}\mathbb{1}\right) \cdots \left(\frac{1}{2}\boldsymbol{\sigma}^{a_2} \cdot \boldsymbol{\sigma}^{b_2} + \frac{1}{2}\mathbb{1}\right)\left(\frac{1}{2}\boldsymbol{\sigma}^{a_1} \cdot \boldsymbol{\sigma}^{b_1} + \frac{1}{2}\mathbb{1}\right). \tag{34}$$

A crucial property of those two group actions is that they commute with each other, i.e., $U\alpha = \alpha U$. One can easily check this for a product state

$$\begin{aligned}
U\alpha(|v_1\rangle \otimes \cdots \otimes |v_n\rangle) &= U(|v_{\alpha^{-1}(1)}\rangle \otimes \cdots \otimes |v_{\alpha^{-1}(n)}\rangle) \\
&= |Uv_{\alpha^{-1}(1)}\rangle \otimes \cdots \otimes |Uv_{\alpha^{-1}(n)}\rangle \\
&= \alpha(|Uv_1\rangle \otimes \cdots \otimes |Uv_n\rangle) \\
&= \alpha U(|v_1\rangle \otimes \cdots \otimes |v_n\rangle),
\end{aligned}$$

which can be extended linearly to all vectors in the space. Thus, it follows that a permutation is an SU(2) equivariant operation. This fact is also the basis of the Schur-Weyl duality we introduce below.

Inspired by Ref. [26], we further consider an operator

$$Q = e^{\sum_{i=1}^{k} c_i \alpha_i} = \sum_{n=0}^{\infty} \frac{1}{n!} \left(\sum_{i=1}^{k} c_i \alpha_i\right)^n, \tag{35}$$

where $c_i \in \mathbb{C}$, which we call generalised permutations. From the expansion, we see that $Q$ also commutes with $U \in$ SU(2), which implies that $Q$ is an SU(2) equivariant operation as well (albeit not unitary, in general). If we further restrict unitarity, i.e., an operator $e^{\sum_i c_i \alpha_i}$ with Hermitian $\sum_i c_i \alpha_i$, such an operator is an element of the set given by Eq. (28).

We now prove the converse of the above statement, which is the main result of this section: All SU(2) equivariant unitary operators can also be written as a form of $\exp[\sum_{i=1}^{k} c_i \alpha_i]$. Even though this can be understood as a consequence of von Neumann's double commutation theorem (see, e.g., Ref. [48]), here we provide constructive proof with a concrete example. The first ingredient for the proof is the *Schur-Weyl duality*.

**Theorem 2** (Schur-Weyl duality). *Under the group actions of $U \in$ SU(2) and the symmetric group $\alpha \in S_n$, the tensor-product space decomposes into a direct sum of tensor products of irreducible modules[6] that determine each other. Precisely, we can write*

$$(\mathbb{C}^2)^{\otimes n} \simeq \bigoplus_D \pi_n^D \otimes J_D \tag{36}$$

*where the summation is over the Young diagram $D$ with n boxes and at most two rows. For each $D$ with $r_1$ boxes in the first row and $r_2$ boxes in the second row, $J_D$ is the irreducible representation of SU(2) with total spin $J = (r_1 - r_2)/2$, and $\pi_n^D$ is the irreducible representation of the symmetric group associated with the given Young diagram $D$.*

We formally introduce the Young diagram and the irreducible representation of $S_n$ in Appendix B. However, for the rest of the discussion in this section, it is fine to skip the details and only consider the dimension of $\pi_n^D$, as we show in the following Corollary.

**Corollary 2.** *From the Schur-Weyl duality, one obtains*

$$(\mathbb{C}^2)^{\otimes n} \simeq \bigoplus_{i=0}^{\lfloor n/2 \rfloor} m_i J_{s_i} \tag{37}$$

*where $m_i$ is the dimension of the irreducible representation of $S_n$ whose Young diagram $D_i$ has $n-i$ boxes in the first row and $i$ boxes in the second row, and $s_i = n/2 - i$ is the total spin.*

*The dimension of the irreducible representation can be computed using the Hook length formula. After some steps, one can obtain*

$$m_i = \begin{cases} 1, & \text{if } i=0, \\ \binom{n}{i} - \binom{n}{i-1}, & \text{otherwise.} \end{cases} \tag{38}$$

We then apply Corollary 1 to this decomposition and obtain all possible SU(2) equivariant gates, given by

$$U = \left\{ \bigoplus_{i=0}^{\lfloor n/2 \rfloor} (U_i \otimes \mathbb{1}_{d_i}) : U_i \in U(m_i) \right\}. \tag{39}$$

In addition, as each $U(m_i)$ has $m_i^2$ independent generators, the total number of parameters is given by

$$\sum_{i=0}^{\lfloor n/2 \rfloor} m_i^2 = \frac{1}{n+1} \binom{2n}{n} \tag{40}$$

Note that Ref. [22] also presents the same result. We also note that, for a quantum gate, we can subtract one from this formula as there is a redundancy for the global phase.

Another ingredient we need is the completeness of the irreducible representation.

---

[6]A vector space where the scalars are a ring.

**Theorem 3** (The density theorem [49]). *Let $V = \mathbb{C}^n$ be an irreducible finite-dimensional representation of a group $G$, i.e., there is a map $R : G \to \mathrm{GL}(\mathbb{C}^n)$. Then $\{R(g) : g \in G\}$ spans $\mathcal{M}_n(\mathbb{C})$.*

See, e.g., Ref. [50] for a proof. The theorem implies that for any $M \in \mathrm{M}_n(\mathbb{C})$, we can find $g_i \in G$ and $c_i \in \mathbb{C}$ such that $M = \sum_{i=1}^k c_i R(g_i)$ when $\mathbb{C}^{\otimes n}$ is the irreducible representation of $G$.

Using the Schur-Weyl duality and the density theorem, we now prove the equivalence between a generalised permutation group action and SU(2) equivariant unitary gates.

**Theorem 4.** *For any* SU(2) *equivariant unitary gate $T$, we can find $c_i \in \mathbb{C}$ and $\alpha_i \in S_n$ such that*

$$T = e^{\sum_{i=1}^k c_i \alpha_i}. \tag{41}$$

*Proof.* First, from Corollary 2, we obtain

$$(\mathbb{C}^2)^{\otimes n} \simeq \bigoplus_{i=0}^{\lfloor N/2 \rfloor} m_i J_{s_i}. \tag{42}$$

Then let $H$ be the generator of $T$, i.e., $T = e^{iH}$ and $H$ is a Hermitian matrix. Looking at Corollary 1, we can move from the description of equivariant unitaries to their generators and see that $H$ can be written as

$$H = \bigoplus_i h_i \otimes \mathbb{1}_{2s_i+1} = \sum_i h_i P_i \tag{43}$$

where $h_i$ is a hermitian matrix in $\mathcal{M}_{m_i}(\mathbb{C})$ and $P_i$ is a projector onto a subspace with total spin $2s_i + 1$. From the density theorem, one can find $\{c_{ij} \in \mathbb{R}\}$ and $\{\alpha_{ij} \in S_n\}$ such that $h_i = \sum_j c_{ij} \alpha_{ij}$ for each $i$. Moreover, each projector $P_i$ can be written as

$$P_i = \prod_{j \neq i} \frac{J^2 - s_j(s_j + 1)}{s_i(s_i + 1) - s_j(s_j + 1)}, \tag{44}$$

where $J = \sum_{i=1}^n \sigma^i/2$ is the total spin operator and $J^2 = J \cdot J$. As $J^2$ has eigenvalues $s_i(s_i+1)$ for each subspace $J_{s_i}$, one can verify that the given operator is indeed a projector. After rewriting

$$J^2 = \frac{1}{4}\Big(3n + \sum_{i \neq j} \sigma^i \cdot \sigma^j\Big) = \frac{4n - n^2}{4} + \sum_{i > j}(i, j) \tag{45}$$

where $(i, j)$ is a transposition, we see that $J^2 \in \mathbb{R}[S_n]$. If we again look at Eq. (43), we can now see that as $h_i, P_i \in \mathbb{R}[S_n]$ our unitary $T = e^{iH}$ is indeed an exponetiated sum of permutations with coefficients in $\mathbb{C}$. $\qquad\square$

## 4.3 Twirling and permutations

In Ref. [24], the Twirling method is proposed to construct an equivariant unitary gate. For a given Hermitian matrix $H$ that is the generator of a unitary gate $V = \exp(iH)$ and a Lie group $\mathcal{G}$, one obtains an equivariant version of it using the twirling formula:

$$\mathcal{T}_U[H] = \int d\mu(g) R(g) H R(g)^\dagger, \tag{46}$$

540  where $\mu(g)$ is the Haar measure for the Lie group $\mathcal{G}$. Then $\mathcal{T}_U[H]$ commutes with any $h \in \mathcal{G}$
541  due to a defining property of the Haar measure, and so does the gate $\exp\{i\mathcal{T}_U[H]\}$.

542      We now show that the twirling formula yields a generalised permutation for $\mathcal{G} = \mathrm{SU}(2)$. For
543  a Hermitian matrix $H \in \mathcal{M}_{2^n}(\mathbb{C})$, we obtain

$$
\begin{aligned}
\mathcal{T}_U[H] &= \int d\mu(g) R(g) H R(g)^\dagger \\
&= \int_U dU\, U^{\otimes n} H (U^\dagger)^{\otimes n} \\
&= \sum_{\sigma, \tau \in S_n} \mathcal{W}g(\sigma^{-1}\tau, d)\mathrm{Tr}[H\tau]\sigma,
\end{aligned}
\tag{47}
$$

544  where $d = 2^n$ is the dimension of the Hilbert space, $\mathcal{W}g(\sigma, d)$ is the Weingarten function, and
545  we identified $\sigma, \tau \in S_n$ as an operator using the representation (see e.g., Refs. [48,51] for the
546  explanation how the last line is obtained). Ultimately, this is a permutation scaled by a real
547  coefficient as required. Furthermore, as $\mathcal{T}_U[H]$ is also Hermitian by definition, we know that
548  $\mathcal{T}_U[H]$ is a Hermitian element of $\mathbb{C}[S_n]$, which can be a generator for an equivariant unitary
549  gate.

550      On the other hand, all generators of equivariant gates can be obtained from the twirling
551  formula. In the spin-basis, we know that each generator of an equivariant gate is given by
552  Eq. (43), i.e., $H \simeq \oplus_i h_i \otimes I_{d_i}$ (where the dimension of $h_i$ and $d_i$ are obtained from the Schur-
553  Weyl duality). As this is an element of the commutant [Eq. (27)], $H$ is also equivariant, i.e.,
554  $HU^{\otimes N} = U^{\otimes N}H$, so $\mathcal{T}_U[H] = H$. In other words, the set of all generators of equivariant gates
555  and the set of all twirled generators are the same:

$$
\begin{aligned}
&\left\{ H \in \mathcal{M}_{2^n}(\mathbb{C}) : U^{\otimes n} e^{iH} = e^{iH} U^{\otimes n} \text{ for all } U \in \mathrm{SU}(2) \text{ and } H = H^\dagger \right\} \\
&= \left\{ \mathcal{T}_U[H] : H \in \mathcal{M}_{2^n}(\mathbb{C}) \text{ and } H^\dagger = H \right\}.
\end{aligned}
\tag{48}
$$

## 4.4  Revisiting three-qubit $\mathrm{SU}(2)$ equivariant gates

557      In this subsection, using the three-qubit vertex gate as an example, we illustrate how to
558  represent our equivariant gates as elements of $\mathbb{C}[S_n]$. We apply Theorem 4 to the three-
559  qubit gate we have found in Sec. 3, using the Schur map given in Eq. (16). A direct con-
560  sequence of the Schur transform is that it defines invariant subspaces under $U^{\otimes 3}$ for any
561  $U \in \mathrm{SU}(2)$, given by $J_{3/2} = \mathrm{span}\{S_3^\dagger|0\rangle, S_3^\dagger|1\rangle, S_3^\dagger|2\rangle, S_3^\dagger|3\rangle\}$, $J_{1/2}^a = \mathrm{span}\{S_3^\dagger|4\rangle, S_3^\dagger|5\rangle\}$, and
562  $J_{1/2}^b = \mathrm{span}\{S_3^\dagger|6\rangle, S_3^\dagger|7\rangle\}$. From the structure of $P(\vec{\theta})$, we know the gate has four generators
563  given by $\{G_I := \mathbf{0}_4 \oplus \mathbb{1}_4, G_X := \mathbf{0}_4 \oplus (X \otimes \mathbb{1}_2), G_Y := \mathbf{0}_4 \oplus (Y \otimes \mathbb{1}_2), G_Z := \mathbf{0}_4 \oplus (Z \otimes \mathbb{1}_2)\}$, where
564  $\mathbf{0}_4$ acts on $J_{3/2}$ whereas $X, Y, Z$ mixes $J_{1/2}^a$ and $J_{1/2}^b$. One can also see that a permutation in $S_3$
565  mixes subspaces $J_{1/2}^a$ and $J_{1/2}^b$ (whereas it acts trivially on $J_{3/2}$ subspace).

566      A matrix representation of a permutation for $\{J_{1/2}^a, J_{1/2}^b\}$ is obtained by applying each per-

mutation to a basis vector, which is given as

$$(1,2) = \begin{pmatrix} 1 & 0 \\ 0 & -1 \end{pmatrix} \otimes \mathbb{1}_2 = Z \otimes \mathbb{1}_2 \tag{49}$$

$$(2,3) = \begin{pmatrix} -1/2 & -\sqrt{3}/2 \\ -\sqrt{3}/2 & 1/2 \end{pmatrix} \otimes \mathbb{1}_2 = -\frac{1}{2} Z \otimes \mathbb{1}_2 - \frac{\sqrt{3}}{2} X \otimes \mathbb{1}_2 \tag{50}$$

$$(1,3) = \begin{pmatrix} -1/2 & \sqrt{3}/2 \\ \sqrt{3}/2 & 1/2 \end{pmatrix} \otimes \mathbb{1}_2 = -\frac{1}{2} Z \otimes \mathbb{1}_2 + \frac{\sqrt{3}}{2} X \otimes \mathbb{1}_2, \tag{51}$$

Each matrix should be read as follows. For example, if we apply $(2,3)$ to $S_3^\dagger |4\rangle$, we have

$$(2,3) S_3^\dagger |4\rangle = -\frac{1}{2} S_3^\dagger |4\rangle - \frac{\sqrt{3}}{2} S_3^\dagger |6\rangle, \tag{52}$$

where the coefficients are from the first column of the matrix representation of $(2,3)$. Note that the permutation transforms $S_3^\dagger |5\rangle$ exactly the same way (but mixes $S_3^\dagger |5\rangle$ and $S_3^\dagger |7\rangle$). Using the above expressions, the remaining elements are obtained as follows (where we dropped $\otimes \mathbb{1}_2$ to simplify the notation):

$$(1,2,3) = (1,2)(2,3) = -\frac{1}{2} \mathbb{1} - i\frac{\sqrt{3}}{2} Y \tag{53}$$

$$(1,3,2) = (1,2)(1,3) = -\frac{1}{2} \mathbb{1} + i\frac{\sqrt{3}}{2} Y. \tag{54}$$

Thus we have

$$I = 1, \qquad\qquad X = -\frac{2}{\sqrt{3}}[(2,3) + 1/2(1,2)] \tag{55}$$

$$Y = i\frac{1}{\sqrt{3}}[2(1,2,3) + 1], \qquad Z = (1,2). \tag{56}$$

However, these operators cannot be generators of our gate as they do not annihilate the $J = 3/2$ subspace (recall that our generators have $\mathbf{0}_4$ on the $J_{3/2}$ subspace). Thus, we need a projector to the $J = 1/2$ subspace, which is given by

$$P_{J=1/2} = \frac{J^2 - 15/4}{3/4 - 15/4} = \frac{5}{4} - \frac{1}{3} J^2 \tag{57}$$

where $J^2$ is

$$J^2 = \frac{1}{4}[\boldsymbol{\sigma}_1 + \boldsymbol{\sigma}_2 + \boldsymbol{\sigma}_3]^2 = \frac{3}{4} + [(1,2) + (2,3) + (1,3)]. \tag{58}$$

By combining the projector and expressions of Pauli operators in $J = 1/2$ subspaces, we can write three generators as

$$G_I = 1 - \frac{1}{3}[(1,2) + (2,3) + (1,3)] \tag{59}$$

$$G_X = -\frac{2}{\sqrt{3}}\Big[-\frac{1}{2} + (2,3) + \frac{1}{2}(1,2) - \frac{1}{2}(1,2,3) - \frac{1}{2}(1,3,2)\Big] \tag{60}$$

$$G_Y = i\frac{1}{\sqrt{3}}\Big[1 + 2(1,2,3) - (1,2) - (2,3) - (1,3)\Big] \tag{61}$$

$$G_Z = (1,2) - \frac{1}{3}[1 + (1,3,2) + (1,2,3)] \tag{62}$$

One can check that each generator annihilates the $J_{3/2}$ subspace (e.g., $G_X |000\rangle = 0$), and acts like a Pauli gate between the $J^a_{1/2}$ and $J^b_{1/2}$ subspaces (e.g., $G_X S^\dagger_3 |5\rangle = S^\dagger_3 |7\rangle$). Also note that, as there is a freedom in choosing two $J = 1/2$ subspaces (any unitary mixtures between $J^a_{1/2}$, $J^b_{1/2}$ are also valid subspaces), the exact form of generators $\{G_I, G_X, G_Y, G_Z\}$ depends on the specific choice of the Schur gate $S_3$ (which is from Eq. (16) for our case).

To summarise, any SU(2) equivariant gate on the three qubit can be written as

$$V(\vec{\theta}) = S^\dagger_3 P(\vec{\theta}) S_3 = \exp\Big[ i \big\{ \theta_0 G_I + \theta_1 G_X + \theta_2 G_Y + \theta_3 G_Z \big\} \Big], \tag{63}$$

which is a generalised permuatation from Eq. (59-62).

We now answer the question raised at the end of the previous section. If we apply our three-qubit gate to the 3rd, 4th, and 7th qubits among eight qubits, we first obtain its representation as a generalised permutation between them and apply it to basis vectors of global spin subspaces. For example, $G_X$ for those qubits are given as

$$G^{(3,4,7)}_X = -\frac{2}{\sqrt{3}}\Big[ -\frac{1}{2} + (4,7) + \frac{1}{2}(3,4) - \frac{1}{2}(3,4,7) - \frac{1}{2}(3,7,4) \Big]. \tag{64}$$

Then, one can construct its matrix form in a certain subspace (e.g., one of the $J_2$ subspaces) by applying it to the basis vectors of the subspace. Then the gate $\exp[-i\theta G^{(3,4,7)}_X]$ can be reconstructed by applying the exponential map.

We finalise this section by introducing an alternative description of these generators using the scalar products. For three operator vectors $\boldsymbol{\sigma}_1, \boldsymbol{\sigma}_2, \boldsymbol{\sigma}_3$, the only possible scalar operators (that are invariant under the group transformation) obtained from those operators are $\boldsymbol{\sigma}_1 \cdot \boldsymbol{\sigma}_2$, $\boldsymbol{\sigma}_2 \cdot \boldsymbol{\sigma}_3$, $\boldsymbol{\sigma}_1 \cdot \boldsymbol{\sigma}_3$, and $\boldsymbol{\sigma}_1 \cdot (\boldsymbol{\sigma}_2 \times \boldsymbol{\sigma}_3)$ up to constant factors, where $A \times B$ is the cross product between two vectors. Thus, another possible representation of a parameterised three-qubit equivariant gate is

$$W = \exp\Big[ i(\theta_{12}\boldsymbol{\sigma}_1 \cdot \boldsymbol{\sigma}_2 + \theta_{23}\boldsymbol{\sigma}_2 \cdot \boldsymbol{\sigma}_3 + \theta_{13}\boldsymbol{\sigma}_1 \cdot \boldsymbol{\sigma}_3) + i\phi\boldsymbol{\sigma}_1 \cdot (\boldsymbol{\sigma}_2 \times \boldsymbol{\sigma}_3) \Big]. \tag{65}$$

Then, it can be shown that this gate is the same as $V(\vec{\theta})$ up to a global phase.

Using

$$\begin{aligned}
(\boldsymbol{\sigma}_1 \cdot \boldsymbol{\sigma}_2)(\boldsymbol{\sigma}_2 \cdot \boldsymbol{\sigma}_3) &= \sum_{a\in\{x,y,z\}}\sum_{c\in\{x,y,z\}} \sigma^a_1 \sigma^a_2 \sigma^c_2 \sigma^c_3 \\
&= \sum_{a\in\{x,y,z\}}\sum_{c\in\{x,y,z\}} \delta_{ac}\sigma^c_1\sigma^c_3 + i\sum_{b\in\{x,y,z\}} \epsilon_{abc}\sigma^a_1\sigma^b_2\sigma^c_3 \\
&= \boldsymbol{\sigma}_1 \cdot \boldsymbol{\sigma}_3 + i\boldsymbol{\sigma}_1 \cdot (\boldsymbol{\sigma}_2 \times \boldsymbol{\sigma}_3), \tag{66}
\end{aligned}$$

and Eq. (33), we obtain

$$2i\boldsymbol{\sigma}_1 \cdot (\boldsymbol{\sigma}_2 \times \boldsymbol{\sigma}_3) = [\boldsymbol{\sigma}_1 \cdot \boldsymbol{\sigma}_2, \boldsymbol{\sigma}_2 \cdot \boldsymbol{\sigma}_3] = [2(1,2)-1, 2(2,3)-1] = 4(1,2,3)-4(1,3,2). \tag{67}$$

In addition, we need another identity $P^2_{J=3/2} = P_{J=3/2}$, which gives

$$(1,2,3) + (1,3,2) = (1,2) + (2,3) + (1,3) - 1. \tag{68}$$

Note that this equality only implies that the LHS and RHS act the same on our vector space. Of course, they are different elements in $\mathbb{C}[S_n]$.

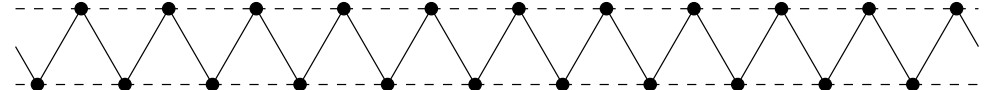

Figure 6: A one-dimensional triangular lattice. We solve the Heisenberg model defined on this lattice using the equivariant gates. The interaction strength between qubits linked with solid lines is given by $J_1$, whereas those between qubits linked with dash lines are $J_2$.

Combining all these together, we can write each generator of $W$ in terms of $\{G_I, G_X, G_Y, G_Z\}$ as

$$\boldsymbol{\sigma}_1 \cdot \boldsymbol{\sigma}_2 = 2(1,2) - 1 = 1 - 2G_I + 2G_Z \tag{69}$$

$$\boldsymbol{\sigma}_2 \cdot \boldsymbol{\sigma}_3 = 2(2,3) - 1 = 1 - 2G_I - \sqrt{3}G_X - G_Z \tag{70}$$

$$\boldsymbol{\sigma}_3 \cdot \boldsymbol{\sigma}_1 = 2(1,3) - 1 = 1 - 2G_I + \sqrt{3}G_X - G_Z \tag{71}$$

$$\boldsymbol{\sigma}_1 \cdot (\boldsymbol{\sigma}_2 \times \boldsymbol{\sigma}_2) = -\frac{i}{2}\big[4(1,2,3) - 4(1,3,2)\big] = -2\sqrt{3}G_Y, \tag{72}$$

which implies that $W$ is just another parameterisation of $V(\vec{\theta})$ (up to a global phase).

## 5 Numerical Simulations

In this section, we numerically demonstrate the efficacy of our equivariant gates for solving quantum many-body Hamiltonians. Our Hamiltonians are Heisenberg models (which are rotationally invariant) defined on frustrated lattices. Even though the Heisenberg models are toy models, they play an important role in understanding the low-temperature physics of some exotic materials [52]. All numerical simulations in this section were performed using the PennyLane [53] software package with the Lightning [54] plugin. Relevant source code is available in GitHub repository [55].

### 5.1 One-dimensional triangular lattice

Let us first consider a one-dimensional triangular lattice given as in Fig. 6. The Hamiltonian we want to solve is

$$H = J_1 \sum_{i=1}^{n}\big[\sigma_i^x \sigma_{i+1}^x + \sigma_i^y \sigma_{i+1}^y + \sigma_i^z \sigma_{i+1}^z\big] + J_2 \sum_{i=1}^{n}\big[\sigma_i^x \sigma_{i+2}^x + \sigma_i^y \sigma_{i+2}^y + \sigma_i^z \sigma_{i+2}^z\big], \tag{73}$$

where we impose the periodic boundary condition $\sigma_{n+1}^{x,y,z} = \sigma_1^{x,y,z}$. Throughout the section, we fix $J_1 = 1$ and consider $J_2 \in \{0, 0.44\}$. When $J_2 = 0$, the Hamiltonian can be transformed into a stoquastic form [56] and a classical algorithm, the variational quantum Monte Carlo (vQMC) with a simple complex-valued restricted Boltzmann machine (RBM), can find the ground state energy extremely accurately [57]. In contrast, such a transformation does not work for $J_2 \gtrsim 0$ [29], and the vQMC with the RBM deviates from the true ground state. We here choose $J_2 = 0.44$ as a recent study [58] reported that such a deviation is maximised near this value. Still, we note that the density matrix renormalisation group can faithfully solve our model as the model is one-dimensional.

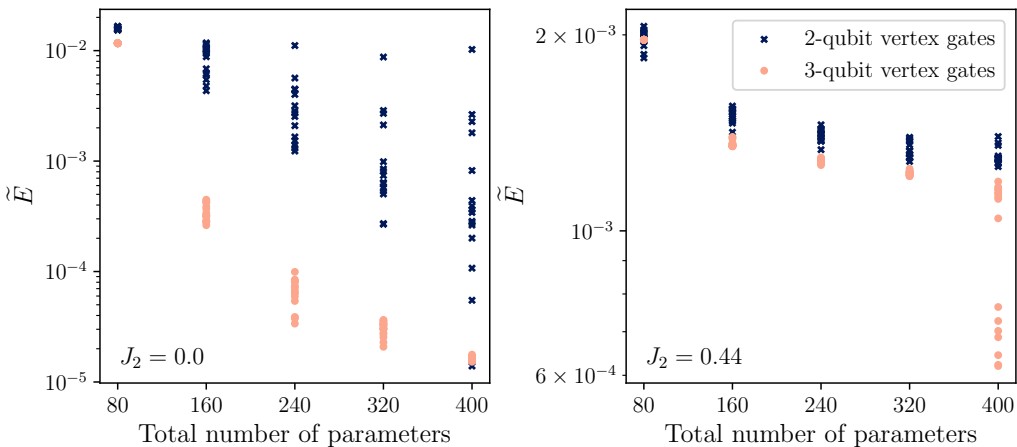

Figure 7: Normalised converged energies as functions of the total number of parameters in a given ansatz for $J_2 = 0.0$ (left) and $J_2 = 0.44$ (right). Each datapoint represents the converged energy obtained from an initial parameter.

We compare the performance of two ansätze for solving this Hamiltonian. The first ansatz only uses the two-qubit vertex gates, which is given by

$$|\psi(\{\theta\})\rangle = \prod_{i=p}^{1}\Bigl[\prod_{j=1}^{n} V_{j,j+2}(\theta_{i,j+n}) \prod_{j=1}^{n/2} V_{2j,2j+1}(\theta_{i,j+n/2}) \prod_{j=1}^{n/2} V_{2j-1,2j}(\theta_{i,j})\Bigr]|\psi_0\rangle, \qquad (74)$$

where $V_{kl}$ is the two-qubit vertex gate acting on $k$-th and $l$-th qubits and $|\psi_0\rangle = (|01\rangle - |10\rangle)^{\otimes n/2}/\sqrt{2}^{n/2}$ is a series of singlets. As $|\psi_0\rangle$ is SU(2) invariant and our circuit is SU(2) equivariant, the output state is also SU(2) invariant. The ansatz has a total of $2np$ parameters, where $p$ is the number of blocks in the ansatz.

Likewise, we also define the second ansatz that consists of the three-qubit vertex gates as

$$|\psi(\{\theta_{i,j}\})\rangle = \prod_{i=p}^{1}\Bigl[\prod_{j=1}^{n} V_{j,j+1,j+2}(\{\theta_{i,4j-3}, \theta_{i,4j-2}, \theta_{i,4j-1}, \theta_{i,4j}\})\Bigr]|\psi_0\rangle, \qquad (75)$$

where $V_{j,j+1,j+2}$ is the three-qubit vertex gate acting on qubits $\{j, j+1, j+2\}$. Also, recall that the three-qubit vertex gate has four parameters, so the ansatz has $4np$ parameters in total.

We now solve the Hamiltonian from Eq. (73) with $n = 20$ for two different values of $J_2 \in \{0.0, 0.44\}$ using the two proposed ansätze by simulating variational quantum eigensolvers (VQEs) using a classical simulator. For each ansatz, we optimise the parameters by minimizing $\langle H \rangle$ using the Adam optimiser. We then compute the converged normalised energies $\widetilde{E} = (\langle H \rangle - E_{\mathrm{GS}})/|E_{\mathrm{GS}}|$ where $E_{\mathrm{GS}}$ is the true ground state energy obtained from exact diagonalisation. We use the number of blocks $p = [2, 4, 6, 8, 10]$ for the ansatz with two-qubit vertex gates. On the other hand, $p = [1, 2, 3, 4, 5]$ is used for the ansatz with three-qubit vertex gates. In addition, inspired by Ref. [59], we initialise the parameters using samples from the distribution $\mathcal{U}_{[0,\alpha]}/(\text{total number of parameters})$ where $\mathcal{U}_{[0,\alpha]}$ is the uniform distribution between 0 and $\alpha$, and $\alpha$ is a hyperparameter giving a relative scaling. We also note that our simulation is performed by computing exact gradients (without shot noise), which is more efficient for classical simulators.

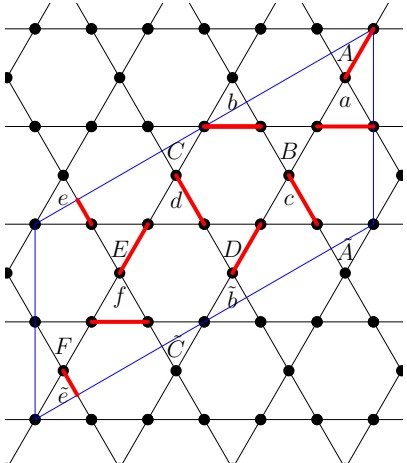

Figure 8: The Kagome lattice. We choose a unit cell with $n = 18$ spins enclosed by blue lines. Red links indicate the singlets which we use as an initial state. Our variational circuit is constructed by applying three-qubit vertex gates to each triangle ($a$-$f$ and $A$-$F$). See the main text for details.

For 16 random initial parameters, we plot the converged normalised energies in Fig. 7 as a function of the total number of parameters. We observe that the converged normalised energies from the ansatz with three-qubit vertex gates are generally closer to the true ground state energy. Especially when $J_2 = 0.44$, the converged energy from the three-qubit vertex gates decreases as the number of parameters increases, whereas that from the two-qubit vertex gates gets flat. This example shows that using a multi-qubit vertex gate is helpful even for solving a Hamiltonian with two-body interactions. We expect this because the circuit ansatz with three-qubit vertex gates is more expressive than two-qubit vertex gates when the same number of parameters is provided.

## 5.2 Kagome lattice

We now extend the previous result to study the model on the Kagome lattice. We consider an $n = 18$ unit cell from the lattice with the periodic boundary condition. Our choice of the unit cell is depicted in Fig. 8.

Formally, the Hamiltonian of the system is written as

$$H = \sum_{\langle i,j \rangle} \left[ X_i X_j + Y_i Y_j + Z_i Z_j \right] \tag{76}$$

where the summation is over all nearest neighbours in the lattice.

We construct an ansatz using three-qubit vertex gates as

$$|\psi(\{\theta_{i,j}\})\rangle = \prod_{i=p}^{1} \prod_{j=A}^{F} V_j(\theta_{i,j}) \prod_{j=a}^{f} V_j(\theta_{i,j})|\psi_0\rangle \tag{77}$$

where $V_{a,\cdots,f}$ ($V_{A,\cdots,F}$) are the three-qubit vertex gates acting on vertices of each triangle $a$ to $f$ ($A$ to $F$, respectively; see Fig. 8). As each block has 12 gates, the total number of parameters is $48p$ (recall that each three-qubit vertex gate has four parameters). We also use a series of

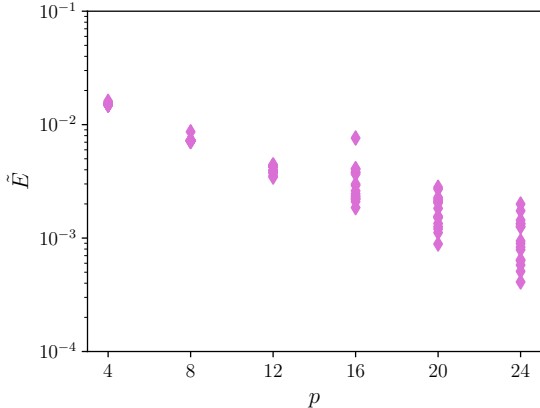

Figure 9: Converged normalised energies as a function of circuit depths for the Heisenberg model on the Kagome lattice. For each value of $p$, 18 random initial parameters are sampled. A full VQE simulation is performed for each random initial parameter, and the converged energy is shown.

singlets as an initial state, where each singlet is indicated by a red link in Fig. 8. Formally, we can write

$$|\psi_0\rangle = \frac{1}{\sqrt{2}^{n/2}} \bigotimes_{\{i,j\}\in S} (|01\rangle - |10\rangle)_{ij}$$

where $S$ is the set of all links.

We numerically optimise the parameters of the circuit by minimizing $\langle H \rangle$. The Adam optimiser is used with the same parameter initialisation techniques as in the previous example. We plot the converged normalised energies as a function of $p$ in Fig. 9. The plot shows that the best-converged energies decrease nearly exponentially with $p$. The smallest converged normalised energy is $\tilde{E} \approx 5.7 \times 10^{-4}$ obtained from $p = 24$, which is comparable to data obtained in Refs. [60, 61] using different ansätze.

To summarise, we have shown that the three-qubit vertex gate introduced in the previous sections is useful for solving the Heisenberg model on different lattices. Given the efficacy of our equivariant gates for solving the ground state problem, we also expect that one can construct a QML model using our gates to classify rotationally invariant datasets such as point clouds [62]. However, as QML models for those datasets without classical pre-processing require a large number of qubits beyond the reach of a classical simulator (which is about $\lesssim 30$ qubits), simulation using a real-world dataset can be considered in future work.

# 6  Connections and discussions

Throughout the previous sections, we have introduced an elegant construction method for SU(2) equivariant quantum circuits based on the Schur transformation. Those circuits can be naturally seen as a spin network, a tensor network of group-invariant tensors. We have further developed a theory of the SU(2) equivariant gates from the Schur-Weyl duality, relating our gates to other known constructions based on the twirling formula and generalised permutations.

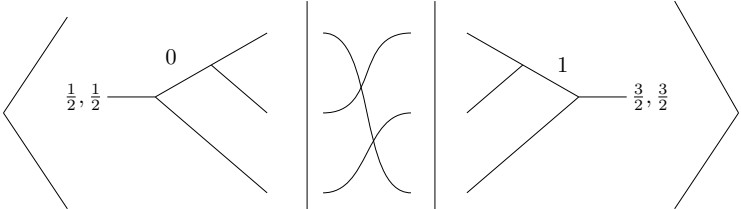

Figure 10: A PQC calculation is an expectation value of a permutation of qubits in the spin-coupling basis.

As spin networks and quantum circuits for permutations appear in lots of different contexts in the field of high energy physics and theoretical quantum computations, we discuss various connections to other fields of research as well as possible future directions of study in the following.

## 6.1   PQC, PQC+, and non-classical heuristic algorithms

The idea of taking spins and coupling them is reminiscent of a computational model already seen in the literature. This idea is at the heart of what we mentioned above and is called permutational quantum computing (PQC), which is centred around the computational class PQC and the closely related PQC+ [20, 31]. This class of problems is important as it provides strong evidence that the transition from permutations to exponentiated sums of the generators of permutations marks a transition to classically hard sampling tasks.

**The PQC model**   In short, PQC is a quantum computing model intimately tied to the structure of a *binary tree* coupling of spins. The original idea stemmed from the notion that spin networks could form a model of quantum computing [43]. To extract a formal computational class from this model, PQC was introduced, which only considers tree-like structures [31]. To achieve this, we take $n$-spins and choose a particular ordering to add the qubits to the already coupled spins (which we can see as a choice of what sequence of spins to apply the $J^2$ operator to). The possible outcomes of this chosen order of spin recoupling, along with the addition of the possible total angular momentum outcomes, give an alternative basis.

PQC is the computational class of problems described as a permutation circuit set between two coupled spin-basis states. Given a permutation operator $U_\sigma$ representing the unitary composed of swap gates implementing the permutation $\sigma \in S_n$, PQC is the set of problems written as:

$$\langle v' | U_\sigma | v \rangle = \langle b' | S^\dagger U_\sigma S | b \rangle \tag{78}$$

where $b$ is some binary label for the computational basis and $S$ is the Schur gate. Schur gate is a core component in PQC because PQC states are simply elements of the spin basis.

The Schur gate is the preparation procedure that sends qubit basis states to spin states. In the PQC literature, these states are often presented by PQC coupling diagrams of the kind seen in Fig. 2. Practically, a standard PQC calculation is merely the inner product between two Schur gates applied to some computational basis states with some SWAP gates in between them. It was shown that this model is, in fact, classically simulable in large part due to the particular tree-like structure of binary spin-recoupling and the restrictions this tacitly forces on the Clebsch-Gordan coefficients dictating their coupling [32]. An immediate observation

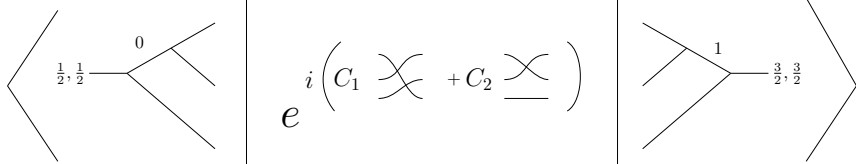

Figure 11: A PQC+ calculation is the exponent of a linear combination of the generators of permutations. Previously, in Fig. 10, the permuted wires stand for the actual permutation, while here they stand for the generators.

we can make, given our above discussion on spin networks, is that PQC diagrams, which we take to be sequentially coupled spin-1/2s, are spin networks with their external wires fixed to specific $J_z$ values. Each PQC basis element is a member of the collection of spin networks of the same tree structure permissible by the recoupling of their spins and a $J_z$ value angular state at the end of the tree.

**PQC+** Despite the initial disappointment that PQC was classically simulable, it has been generalised to a broader model that is believed to be unlikely to have this property. The extended model is known as PQC+ where instead of working with a permutation $\sigma \in S_n$, we work with unitaries generated by sums of elements of the permutation algebra $\mathbb{C}[S_n]$: this is composed of elements $f = \sum_i c_i \sigma_i$ with $U_f = e^{if} = e^{i\sum_k c_k \sigma_k}$ so, in the end, computations are defined in the following manner:

$$\langle v' | U_f | v \rangle. \tag{79}$$

As was mentioned above, the belief in the resilience of this model to 'dequantisation'[7] rests on the fact that PQC+ is capable of approximately computing unitary $S_n$ Fourier coefficients in polynomial time; the details can be found in Ref. [26]. The general idea is that, much like in a traditional Fourier transform, to calculate the Fourier coefficient of any element, one must get the component from every element in the original basis, so in the worst case, one must go through as many components as there are basis elements classically. For an $S_n$ Fourier transform, there are a permutational number of elements[8], as such even an approximate classical polynomial time algorithm to compute the worst case is unlikely. This property relates to claims of super-exponential speed-up as permutational complexity grows considerably faster than the exponential. For more details, we direct the reader to Refs. [20, 26], where one also finds some practical application in condensed matter calculations in accessing coefficients relevant to the Heisenberg chain.

**Spin-network circuits as *non-classical heuristics*** The major observation in work on PQC+ is that, for a Hamiltonian $H = \sum_i c_i \sigma_i$, we can approximate $\langle u| \exp(-itH)|v\rangle$ in polynomial time using a quantum circuit. As the Hamiltonian is in the space $\mathbb{C}[S_n]$ (the algebra of permutations), we are computing $\mathbb{C}[S_n]$ Fourier coefficients in polynomial time. Given that this computation of a Fourier coefficient using the best-known classical algorithm requires one to run over all of $S_n$ that is super-exponential in size, PQC+ allows a super-exponential speed-up. This suggests that, in general, elements of the form $\langle u|e^{i\sum_k c_k \sigma_k}|v\rangle$ cannot be efficiently classically computed [20]. These elements, however, are exactly the form of our parameterised ver-

---

[7]Quantum computing shorthand for the situation where a quantum algorithm is proposed to be faster than possible classical alternatives only for a new classical method to be devised that eliminates this speed-up.

[8]This is sloppy, as one actually runs over the number of irreps which slightly smaller than permutationally, i.e., factorially, large but remains super-exponential.

755 tex gates – this tells us that the paths through parameter space our vertex gates move through
756 are classically inaccessible. This motivates us to introduce the term *Non-classical heuristics*
757 – parameterised ansätze that are defined as moving through spaces that cannot be accessed
758 classically in polynomial time. However, we should note that this idea does not tell us if mov-
759 ing through this space is useful; the space may still be barren [63]. We have shown that the
760 form of these problems matches those of SU(2) (perhaps more generally SU($d$)) equivariant
761 gates, which are of direct practical interest. The principle, then, is that there could be practical
762 problems, such as SU(2) equivariant optimisation problems, for which we can design quantum
763 circuit heuristics, such as spin-network circuits, that cannot be replicated classically because
764 the maps they implement cannot be replicated in polynomial time.

765     In terms of the approaches to machine learning presented in PQC+ to date and our spin-
766 network circuits, it should be noted that there is a technical distinction between the methods
767 used. The PQC+ focuses on tuning the coefficients $c_i$ of the exponent $\sum_i c_i \sigma_i \in \mathbb{C}[S_n]$. In our
768 spin-network circuits, we parameterise the SU(2) distinguishable spin-spaces and mix spin
769 irreps of the same $J$-value in the Schur-Weyl decomposition via unitaries (see Corollary 1).
770 Though both exist in the same space, the way in which one moves through that parameter
771 space is very different.

## 6.2  Further directions

773 **Mixed valency networks**  In this work, we have focused on the traditional spin network
774 perspective, where the same valency exists throughout the graph. In the usual contexts for spin
775 networks, there is a physical motivation for this (see Appendix C). However, from a quantum
776 algorithms perspective, there is no fundamental reason not to mix the valencies. While it is
777 true that larger vertex maps are likely more expressive than small ones as they are generated
778 by a larger set of permutations, it could also be possible that an architecture with small vertex
779 ones is advantageous for practical training.

780 *G*-**Networks**  The idea of graphs with edges indexed by representations of a group in the
781 manner presented here is more general than SU(2). The most obvious extension is to SU($N$),
782 for which many of the technical elements used in the SU(2) still remain. In particular, we
783 have generalised Clebsch-Gordan coefficients. Thus, we can still decompose products of irreps
784 into block diagonal form, allowing us to express the idea of coupling two representations and
785 presenting this as a collection of irrep indexed diagrams. These can then be parameterised
786 in the manner used throughout this paper to create general parameterised equivariant maps
787 suitable for machine learning. In the specific case of SU($N$), there is reason to believe that the
788 same hopes of finding algorithms particularly suited to quantum computing remain, namely
789 because the speed-up arguments presented in Ref. [20] apply to SU($N$). From an applications
790 perspective, this would allow for this research to connect to condensed matter physics, which
791 would be an excellent candidate domain for such non-classical heuristic algorithms [64–66].
792 Leaving SU(2) for higher dimensions, however, is not without complications. One striking dif-
793 ference is that while with SU(2) we have one irreducible representation per dimension, the size
794 of which identifies the representation, for SU($N$) the irreps are identified by 'highest weights'
795 which are $N-1$ (half) integers that provide representations only in certain dimensions. While
796 this may be surmountable, it is likely that general *G*-networks will be markedly more complex
797 than spin networks.

798     Implicitly, we are relying on the ability to construct all representations from irreducible

ones, which tells us that our groups of interest will typically also need a notion of compactness or that the situation of interest is restricted to elements where irreducible deconstruction can be relied upon. Without this guarantee, we cannot expect that it is enough to identify a structure of irreducible representations to construct the other representations. An interesting perspective on this direction is that it can be seen as fusing the perspective of equivariant QML algorithms with work done in tensor networks. Indeed, a spin network is essentially a tensor network decomposition of some map where the tensors involved are always SU(2) invariant. The general version of this through *G*-networks is essentially tensor network decomposition of *G*-equivariant maps into *G*-invariant 'harmonic' tensors.

**Quantum Gravity**    While the connection to the Loop Quantum Gravity (LQG) has only been indirectly alluded to in this work, it holds a natural significance. In LQG, space itself is a quantum state on which geometric operators act to give values for length, area, angle, and volume. The basis of its state space is made up of spin networks. A more detailed explanation of the LQG can be found in Appendix C.

As with all theories of quantum gravity, LQG faces a general lack of decisive experimental data. However, our research demonstrates that quantum computing can potentially represent some fundamental mathematical structures that underlie the quantised nature of space in LQG. This opens up the possibility of exploring these structures numerically using quantum computing devices.

While in LQG, the dynamics of spin networks often involve broader groups such as SL(2; $\mathbb{C}$) that correspond to relativistic symmetries, we still find value in the SU(2) (Euclidean) models. This is because even in the most developed LQG models addressing quantised relativistic space-time, the states of space themselves are still projected onto SU(2) [67]. In summary, though tackling the full dynamics directly might prove challenging through this approach, exploring the kinematic aspects is well within reach. Interestingly, the PQC literature already contains the treatment of a limited class of spin networks to calculate the Ponzano-Regge amplitudes [31], which are the transition amplitudes for the topological quantum field theory known as the Ponzano-Regge model, which itself is studied as a model for quantum gravity [68]. In this context, spin networks are not viewed as states but as transition maps in a 2+1 Euclidean gravity setup, i.e., non-relativistic dynamics over lower dimensions (see Appendix C for details). While there might be an absence of the full group of relativistic symmetries, investigating even a simplification of these transition amplitudes and the associated objects, termed spinfoams, could yield valuable insights.

In a different context, an additional observation mentioned above is the possibility of generalising the models we have explored. This includes considering networks with mixed vertices or looking into groups like SU($N$), which extend beyond what is typically seen in LQG. Indeed, in LQG, even models with vertices larger than four are considered exotic. The exploration of the properties of this wider class of models could prove useful in quantised gravity. Such generalisation would be in the spirit of the work on probabilistic theories [69,70]. Those studies often consider a diverse landscape of theories similar to quantum mechanics to discover why quantum mechanics, particularly, is seen in nature. Investigations of different valency spin networks could proceed along similar lines.

# 7 Conclusion

In this paper, we have put forward a theoretically motivated ansätze based on spin networks, a form of SU(2) equivariant tensor networks. This offers a way to design SU(2) equivariant variational quantum algorithms, which are natural for rotationally invariant quantum systems, based on the Schur map induced by a spin-coupling diagram. Furthermore, we show that our approach leads to the same parameter spaces as generated by the twirling formula but in a direct manner that avoids the twirling computation for many-qubit gates, which is highly non-trivial. For the two and three-qubit gate cases, we further justify our approach with numerical results solving the ground state problem of the SU(2) symmetric Heisenberg models on the one-dimensional triangular lattice and on the Kagome lattice. Connecting to the broader literature, we also show that SU(2) equivariant gates are identical to the generalised permutations discussed in the context of PQC+ [26].

The connection to PQC+ is also used to argue how our ansätze moves through a parameter space that a classical algorithm finds difficult to access. The original observation in Ref. [26] showed that the expectation value of generalised permutations in the spin-basis calculates $S_n$ Fourier coefficients in polynomial time (a possible super-exponential speed-up) and our work now extends this to SU(2) equivariant gates. This leads to our introduction of the term *non-classical heuristics* for quantum variational techniques, which can be argued to access regions of the parameter space that are classically intractable.

It is our hope that future research in this direction can extend this notion to rigorous complexity arguments by finding a task with SU(2) symmetry that is solvable by SU(2) equivariant circuits where no known efficient classical algorithm exists. For example, Ref. [71] has proven quantum advantage in an ML task by designing a dataset whose classification task is convertible to the discrete logarithm problem, which is efficiently solvable by a QML algorithm, yet an efficient classical algorithm is deemed impossible (unless discrete logarithm problem is in BPP). Similarly, we expect it is possible to design an ML task related to the Fourier transformation over $S_n$, also establishing rigorous quantum advantage arguments in this domain.

## Acknowledgements

RDPE and CYP contributed equally to this work. RDPE would like to acknowledge useful conversation and comments from Sergii Strelchuk, Deepak Vaid, and Pierre Martin-Dussaud. CYP thanks Seongwook Shin for his helpful comments. All authors thank the Xanadu QML and software research teams, with special thanks to Maria Schuld, David Wierichs, Joseph Bowles, and David Wakeham. This research used resources of the National Energy Research Scientific Computing Center, which is supported by the Office of Science of the U.S. Department of Energy under Contract No. DE-AC02-05CH11231.

# A Formal introduction to spin networks

Despite having a modest presentation, the gate architectures seen in Sec. 3 cannot be understood beyond a superficial depth without grasping the motivating concept of the spin network

more deeply. The spin network can be seen as a type of tensor network where the vertices are invariant under SU(2) actions, and the contraction edges are indexed by irreps of SU(2). This relates to a particular representation of equivariant linear maps as *harmonic tensor networks*[9] over SU(2), by which we mean a tensor network where the tensors involved are all equivariant with respect to the given group. Here, however, we will give the classical presentation of a spin network as a labelled graph in order to allow the interested reader to follow the spin network literature more easily.

**Labelled Directed Graphs.**    A spin network is a particular form of a labelled directed graph. A directed graph $\Gamma$ is an ordered pair $\Gamma = (\mathcal{N}, \mathcal{L})$, where $\mathcal{N} = \{n_1, \ldots, n_N\}$ is a finite set of $N$ nodes, and $\mathcal{L} = \{l_1, \ldots, l_L\}$ a finite set of L edges (traditionally referred to as links in the Loop quantum gravity literature), endowed with a target map $t : \mathcal{L} \to \mathcal{N}$ and a source map $s: \mathcal{L} \to \mathcal{N}$, assigning each edge to its end and start points respectively. We denote $\mathcal{S}(n)$ (respectively $\mathcal{T}(n)$) the set of edges for which the node $n$ is the source (respectively the target). The valency of a node $n$ is the number of edges with $n$ as an endpoint, i.e., $|\mathcal{T}(n)|$. A graph is said to be $p$-valent if the valency of each node is $p$.

**Intertwiners.**    Before defining spin networks proper by restricting ourselves to labelled directed graphs of a certain type, it will be profitable to define the concept of intertwiners. Let us say that we have two vector spaces $V$ and $W$ on which we have representations $U_V, U_W : G \to V$ of a group made up of elements $g \in G$ and its algebra $\mathfrak{g}$, an intertwiner is a linear map $T : V \to W$ which satisfies:

$$T(U_V(g) \circ v) = U_W(g) \circ T(v) \tag{A.1}$$

where $v \in V$. This is alternatively characterised by the commuting diagram:

$$
\begin{array}{ccc}
V & \xrightarrow{\;T\;} & W \\
{\scriptstyle U_V(g)}\big\downarrow & & \big\downarrow{\scriptstyle U_W(g)} \\
V & \xrightarrow{\;T\;} & W.
\end{array}
\tag{A.2}
$$

This shows us that an intertwiner is an equivariant map. This is also referred to as a covariant map, depending on the literature.

The space of intertwiners denoted $\mathrm{Hom}_G(V, W)$, is a subspace of the vector space of linear maps $\mathrm{Hom}(V, W)$ from $V$ to $W$. Given a space of equivariant maps under the group $G$ we can make the following useful identification of the *equivariant* maps with an isomorphic space of *invariant* states

$$\mathrm{Hom}_G(V, W) \cong \mathrm{Inv}_G(V \otimes W^*), \tag{A.3}$$

where $W^*$ is the dual space of $W$ made up of maps from $W$ to the complex numbers. Here we define an invariant space as

$$\mathrm{Inv}_G(E) \overset{\mathrm{def}}{=} \{\psi \in E \mid \forall g \in G, g \cdot \psi = \psi\}. \tag{A.4}$$

We can see by the construction from $G$ equivariant maps that the states in $E$ when acted on by by $G$ via the representation $U_V \otimes U_W^{\dagger}$ must be such that for any $v \otimes w^{\dagger} \in V \otimes W^*$

$$(U_V \otimes U_W^{\dagger}) v \otimes w^{\dagger} = (Id \otimes U_W U_W^{\dagger}) v \otimes w^{\dagger} = v \otimes w^{\dagger} \tag{A.5}$$

---

[9]Harmonic in the sense of harmonic analysis and decomposition of functions over representations, see Ref. [67].

910    which is the source of their invariance.

911    Let us consider again the directed graph $\Gamma$. We denote $\Lambda_\Gamma$ by the set of labellings $j$ that
912    assign to any edge $l \in \mathcal{L}$ an SU(2) irreducible representation characterised by the spin number
913    $j_l \in \mathbb{N}/2$. Given a labelling $j \in \Lambda_\Gamma$, we write

$$\mathrm{Inv}(n, j) \overset{\text{def}}{=} \mathrm{Inv}_{\mathrm{SU}(2)} \left( \bigotimes_{l \in \mathcal{S}(n)} V_{j_l} \otimes \bigotimes_{l \in \mathcal{T}(n)} V_{j_l}^* \right), \tag{A.6}$$

914    where the $V_{j_l}$ are the spaces of the irreps $j_l$ associated with the edges. Using the concept of
915    invariant subspace, we can now define a spin network.

**spin networks.**    A *spin network* is a triple $\Sigma = (\Gamma, j, \iota)$, where $\Gamma$ is a directed graph, with
917    a labelling $j \in \Lambda_\Gamma$ on its edges, and a map $\iota$ that assigns to every $n \in \mathcal{N}$ an intertwiner
918    $|l_n\rangle \in \mathrm{Inv}(n, j)$.

**Clebsch-Gordan coefficients and the vertex basis**    Having described the spin network ab-
920    stractly, it can be practical to choose a specific basis in order to look at how the vertices are
921    represented as matrices. The smallest possible non-trivial intertwiner is three-valent, and we
922    shall see that we can construct all larger valences from these. For the three-valent intertwiner
923    the space is $\mathrm{Inv}_{\mathrm{SU}(2)} \left( J_{j_1} \otimes J_{j_2} \otimes J_{j_3} \right)$[10] and it can be given a basis by sequentially coupling the
924    first two spins and then contracting the result with the third. Firstly, we need to map the tensor
925    product of the first two spins $J_{j_1} \otimes J_{j_2}$ to the direct sum basis $J_{j_1} \oplus J_{j_2}$ as in

$$J_{j_1} \otimes J_{j_2} \simeq \bigoplus_{k=|j_1-j_2|}^{j_1+j_2} J_k \tag{A.7}$$

926    Here the equivalence is given by the intertwiner map:

$$\iota \left\{ \begin{array}{l} J_{j_1} \otimes J_{j_2} \to \bigoplus_{k=|j_1-j_2|} J_k \\ |j_1, j_2; m_1 m_2\rangle \to |km\rangle \end{array} \right. \tag{A.8}$$

927    Written in this form, we can see that the intertwiner map is a change of basis to block diag-
928    onalising the representation, and each block is an irreducible representation. This is just the
929    Schur map when we have qubits, i.e., spin-1/2s as the first two spaces. The matrix coefficients
930    of the map $\iota$ are given by the Clebsch-Gordan coefficients[11]

$$C_{j_1 m_1 j_2 m_2}^{jm} := \langle j_1 m_1; j_2 m_2 | jm \rangle \tag{A.9}$$

931    Clebsch-Gordan coefficients are usually first encountered by physicists during undergraduate
932    courses in atomic physics. They are typically presented as the obscure coefficients that dictate

---

[10]Note we have dropped the reference to the last space being the conjugate, this is common in the literature as
they are isomorphic.

[11]In the spin network literature, we often see that vertices are described via Wigner symbols instead of Clebsch-
Gordan coefficients as seen here. The Wigner symbols are an equivalent way to decompose three vector spaces as
is done here, which is more symmetric. Since we are looking to derive computations with well-defined input and
output, it is simpler to use this basis instead. See Ref. [38] for more details.

how different (atomic) spin states $|j_1, m_1\rangle$ and $|j_2, m_2\rangle$ combine together to form a combined $|j, m\rangle$ state as seen in the equation:

$$|jm\rangle = \sum_{m_1=-j_1}^{j_1} \sum_{m_2=-j_2}^{j_2} c_{j_1,j_2,m_1,m_2}^{jm} |j_1 m_1 j_2 m_2\rangle, \tag{A.10}$$

where the coefficients are taken to be non-zero only when the Clebsch-Gordan conditions hold:

$$\begin{aligned} j_1 + j_2 + j &\in \mathbb{N} \\ |j_1 - j_2| \leqslant j &\leqslant j_1 + j_2. \end{aligned} \tag{A.11}$$

Notably, the space $\mathrm{Inv}_{\mathrm{SU}(2)}\left(J_{j_1} \otimes J_{j_2} \otimes J_{j_3}\right)$ is one dimensional, meaning there is only one intertwiner up to a scalar. This makes sense because, in the three-valent case, the choice of two spins completely fixes the third [67].

For higher valence networks, we can build a similar basis by reapplying the decomposition procedure seen in Eq. (A.7) until all the tensor products are replaced by direct sums. For example, in the case of four-valent spin networks, we reapply Eq. (A.7) to three-valent product spaces tensored with the third spin

$$\left(\bigoplus_{k=|j_1-j_2|}^{j_1+j_2} J_k\right) \otimes J_{j_3} = \bigoplus_{j_{12}=|j_1-j_2|}^{j_1+j_2} \bigoplus_{k=|j_{12}-j_3|}^{j_{12}+j_3} J_k. \tag{A.12}$$

This, in terms of states and Clebsch-Gordan coefficients, leads to the following:

$$|(j_1 j_2)j_3; j_{12} kn\rangle = \sum_{m_1,m_2,m_3,m_{12}} C_{j_1 m_1 j_2 m_2}^{j_{12} m_{12}} C_{j_{12} m_{12} j_3 m_3}^{kn} \bigotimes_{i=1}^{3} |j_i, \mathrm{m}_i\rangle. \tag{A.13}$$

It is important to note that there is freedom in ordering the breakdown of a tensor product of three elements into direct sums. Here, we take the first two spins, consider the resultant direct sum, and then take the tensor product with the third space. This could be reversed, and we could take the second and third or the first and third. These separate decompositions amount to different basis choices which play a role in the structure of permutational quantum computing discussed above (see Sec. 6). The quantum gravity community is mostly interested in three- and four-valent spin networks due to a relationship with 2D and 3D space models of gravity (see further below in this section and Refs. [67, 72]). Our interests are, in principle, broader than this, though all spin networks can be decomposed into three-valent ones. In addition, there is also a direct relationship with the present quantum computing literature and three-valent intertwiners due to the work on PQC.

# B  The representation theory of the symmetric group

In this Appendix, we briefly introduce the irreducible representation of the symmetric group $S_n$.

Consider a partition of a positive integer $n$ to be a monotonically decreasing sequence of positive integers, $\lambda = (\lambda_1, \lambda_2, \cdots)$ that sum to $n$. We can associate these with cycle shapes

960 of $S_n$. For example, given ten elements, we can associate the partition $\lambda = (4, 2, 2, 2)$ with a
961 permutation decomposable into one four-cycle and three two-cycles.

962 A Young diagram is a diagrammatic depiction of the cycle shapes of $S_n$. Typically, the largest
963 cycle goes at the top, and for every element in the cycle, we add a box, as seen here:

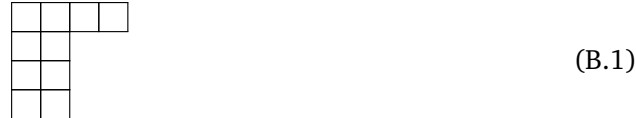

$$(B.1)$$

964 A *Standard filling* of a Young diagram is a bijective map of the numbers from 1 to $n$, where
965 n is the number of boxes such that the entries increase along the rows and down the columns.
966 The standard filled Young diagram is called a *Young tableau*

$$
\begin{array}{|c|c|c|c|}
\hline 1 & 3 & 4 & 7 \\
\hline 2 & 8 \\
\cline{1-2}
5 & 9 \\
\cline{1-2}
6 & 10 \\
\cline{1-2}
\end{array}
\tag{B.2}
$$

967 We can act with an element of the symmetric group on the tableau by simply applying the
968 permutation $\alpha \in S_n$ to the filling numbers.

969 Let us define the equivalence class $R(T)$ of permutations that only move elements about
970 *within their rows*. In this way, we define the row stabilisers, simply the product subgroups
971 $\bigotimes_{p \in \lambda} S_p$. In our earlier example, it would be the space $S_4 \otimes S_2 \otimes S_2 \otimes S_2$. Analogously, we can
972 also describe the column stabilisers $C(T)$.

973 To describe the irreps of $S_n$, we will need the Young polytabloid:

$$
e_T = \{T\} = \sum_{\alpha \in C(T)} \text{sgn}(\alpha) \alpha \triangleright T
\tag{B.3}
$$

974 where $\text{sgn}(\alpha)$ is the parity function giving 1 for an even permutation or $-1$ for an odd one.
975 We note that $\alpha \triangleright T$ is not necessarily a Young tableau due to its non-standard filling.

976 For example, given the tableau

$$
\begin{array}{|c|c|}
\hline 1 & 2 \\
\hline 3 \\
\cline{1-1}
\end{array}
\tag{B.4}
$$

977 the polytabloid is given by

$$
\left\{ \begin{array}{|c|c|}\hline 1 & 2 \\\hline 3 \\\cline{1-1}\end{array} \right\} = \text{sgn}(Id)\begin{array}{|c|c|}\hline 1 & 2 \\\hline 3 \\\cline{1-1}\end{array} + \text{sgn}((1,3))\begin{array}{|c|c|}\hline 3 & 2 \\\hline 1 \\\cline{1-1}\end{array} = \begin{array}{|c|c|}\hline 1 & 2 \\\hline 3 \\\cline{1-1}\end{array} - \begin{array}{|c|c|}\hline 3 & 2 \\\hline 1 \\\cline{1-1}\end{array}
\tag{B.5}
$$

978 A Specht module is a module[12] spanned by polytabloids $e_T$ where $T$ is the index correspond-
979 ing to all tableaux of shape $\lambda$. That is to say.

$$
Sp^{(\lambda)} = \{c_1 e_{T_1} + c_2 e_{T_2} + c_3 e_{T_3} + \cdots | c_1, c_2 \ldots \in \mathbb{C} \text{ and } T_1, T_2 \ldots \text{ are tableaux of shape } \lambda\}.
\tag{B.6}
$$

---

[12]A generalisation of a vector space. A vector space has scalars belonging to a field. Still, a module has scalars
from a ring (meaning the multiplicative operation does not have to be a commutative group). Though we range
over the field $\mathbb{C}$ in our example, this is not generally the case, hence the literature using the term module.

It can be shown that the Specht modules are the irreps of $S_n$ [73].

In the context of the above work, let $n = 3$, and restrict to the Young diagrams with at most two rows which correspond to the multiplicity of elements of SU(2) by Schur-Weyl duality. These are

$$\boxed{\phantom{1}}\boxed{\phantom{2}}\boxed{\phantom{3}} \qquad \text{and} \qquad \begin{array}{c}\boxed{\phantom{1}}\boxed{\phantom{2}}\\\boxed{\phantom{3}}\end{array}. \tag{B.7}$$

The irreducible representations of $S_3$ associated with the first diagram are dimension 1, and the second diagram is dimension 2. More precisely, the Specht module for the first diagram is generated by a single vector:

$$\left\{\boxed{1\,|\,2\,|\,3}\right\} = \boxed{1\,|\,2\,|\,3}. \tag{B.8}$$

For the second diagram, it is generated by two vectors which correspond to the two possible tableau

$$\left\{\begin{array}{c}\boxed{1\,|\,3}\\\boxed{2}\end{array}\right\} = \begin{array}{c}\boxed{1\,|\,3}\\\boxed{2}\end{array} + \begin{array}{c}\boxed{2\,|\,3}\\\boxed{1}\end{array} \tag{B.9}$$

and

$$\left\{\begin{array}{c}\boxed{1\,|\,2}\\\boxed{3}\end{array}\right\} = \begin{array}{c}\boxed{1\,|\,2}\\\boxed{3}\end{array} - \begin{array}{c}\boxed{3\,|\,2}\\\boxed{1}\end{array}. \tag{B.10}$$

Referring back to the Schur-Weyl decomposition where the irreps of $S_n$ give the multiplicities of the SU(2) irreps, we observe:

$$(\mathbb{C}^2)^{\otimes 3} \simeq J_{3/2} \oplus 2J_{1/2}, \tag{B.11}$$

as the three-row element corresponds to the fully symmetric subspace of the three-qubit components, i.e., spin-$\frac{3}{2}$ and the mixed representation corresponds to the spin-$\frac{1}{2}$. For more details, see Refs. [47, 74].

# C   LQG, quantised geometry, and the geometry of SU(2) equivariant algorithms

## C.1   What is LQG?

In this appendix, we refer to the work done in Ref. [72] for more details. Loop quantum gravity (LQG) is based on the idea that space-time is quantised, and it describes space using a Hilbert space whose basis is indexed by *spin networks*. These spin networks can be seen as the dual space of tessellating simplices, such as triangles in 2+1 dimensions or tetrahedra in 3+1 dimensions. Length, angle, area, and volume operators act on these spin networks, yielding quantised answers. The dynamics of LQG are described by spinfoams, which can be viewed as maps between spin networks. Spinfoams are the fundamental objects, and spin networks can be seen as particular foliations of the spinfoams, where each 'moment' is a superposition of states of quantised space represented by the spin networks. The transition amplitudes are

obtained by summing over all spinfoams that are bounded by the initial and final spin networks that are being transitioned between.

LQG's historical development has been involved, and although more elegant routes to LQG may emerge if the theory proves successful, we currently rely on the present understanding. Given the theory's novelty to some readers, we provide a brief outline of how one arrives at spin networks and spinfoams. General relativity is typically modelled by a manifold $\mathcal{M}$ with a metric $g_{\mu\nu}$ that varies from point to point. To quantise gravity via second quantisation, a time parameter is needed. This can be achieved by ADM splitting [72], which divides the space into 3D foliations $\Sigma_t$ indexed by $t \in \mathbb{R}$, making space-time a product of $\Sigma$ and $\mathbb{R}$. The classical configuration space $\mathcal{C}$ is defined by possible metrics $q_{ab}$ on 3D foliations $\Sigma_t$, and the Einstein equations govern how we move from one slice with metric $q_{ab}$ to another. One can go on to define an extrinsic curvature $k_{ab}$, which defines a 'momentum' on $\Sigma_0$. Together with $q_{ab}$ they describe a classical state of space-time and define a point in the phase space $\mathcal{P}$.

Diffeomorphism invariance imposes constraints on the phase space, indicating that only a subspace of $\mathcal{P}$ is needed to describe physical states. To quantise, we move from phase space $\mathcal{P}$ to a Hilbert space $\mathcal{H}$, and the coordinates of $\mathcal{P}$ become operators on $\mathcal{H}$. Though it should be noted on the way Ashtekar-Barbero variables $(A_i^a, E_a^i)$ are used instead of $(q_{ab}, k_{ab})$, which brings general relativity closer to successfully quantised gauge theories. Truncation is performed by taking a finite graph $\Gamma$ embedded within $\Sigma$, reducing the phase space from 3D to 1D. Holonomies along the links of $\Gamma$ are used to describe the relevant parts of the phase space, resulting in a finite-dimensional space. The Hilbert space $\mathcal{H}_\Gamma$ is a space of square-integrable functions of the holonomies.

There are other constraints in LQG, notably the Gauss constraint, which restricts the Hilbert space to the invariant subspaces. This in turn leads to the final Hilbert space in LQG being a sum over all possible SU(2) invariant graphs, where each graph represents a spin network with an edge label as irreducible representations of SU(2) and vertices as intertwiners of the attached edges. These spin networks then form a basis for describing quantum states of space in LQG, and indeed, as is discussed below, they have an interpretation in terms of quantised shapes with appropriate operators.

## C.2  Seeing geometry in spin networks

It is possible to view SU(2) coupling theory, typically understood through the arcane use of Clebsch-Gordan coefficients or alternatively by Wigner or Racah symbols, as statements about geometries with quantised values. While this approach is presently unusual, it is more intuitive. This is the source of the geometric interpretation of spin networks.

**The quantised triangle perspective**  The Clebsch-Gordan conditions are more interesting than they appear. Consider them once more:

$$
\begin{aligned}
j_1 + j_2 + j &\in \mathbb{N} \\
|j_1 - j_2| \leqslant j &\leqslant j_1 + j_2.
\end{aligned}
\tag{C.1}
$$

The reason they are more interesting than they seem at first sight is hinted at by the specific name for the second of these constraints. It is known as the *triangle inequality*. Given a triangle with sides with lengths that we will suggestively label $j, j_1$ and $j_2$, which are half integers (i.e., in $\mathbb{N}/2$), it is an elementary fact that the length $j$ in a valid triangle must be smaller

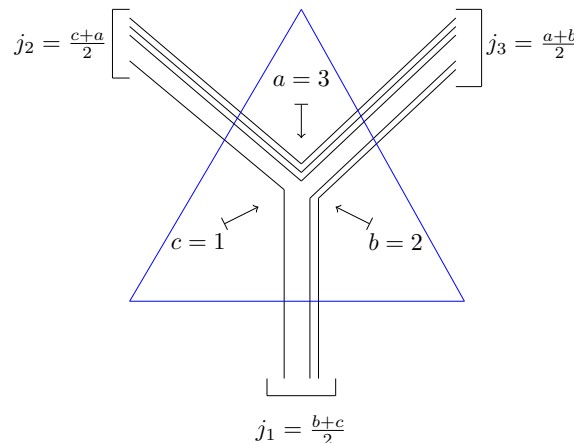

Figure 12: Relationship between the CG coefficients for discretised edge lengths and the non-negative integers $a, b$, and $c$. We can see these as indicating pairings of a decomposition of the edge lengths in amounts of $\frac{1}{2}$ [72].

than or equal to the combined lengths of the other sides and larger than the magnitude of the difference of the other edges[13]. This invites the interpretation of non-zero spin-coupling coefficients as indicating the existence of valid triangles with spin magnitude edge lengths. The first condition is a little more mysterious. The condition that the three half-integers sum to a whole number implicitly requires that the number of summed $\frac{1}{2}$s is even. If we recall however that spin-$\frac{n}{2}$ is the symmetric subspace of $n$ copies of spin-$\frac{1}{2}$, we can interpret this as demanding that, when decomposed into spin-$\frac{1}{2}$ components, there are enough spin-$\frac{1}{2}$s to be paired up. This perspective is further justified in that both conditions can be rewritten as $2j_1 = b + c$, $2j_2 = c + a$, $2j_3 = a + b$ for three non-negative integers $a, b$, and $c$. This permits us to understand both conditions in terms of the picture seen in Fig. 12, which matches these conditions on triangles to the ability to bring three half-integer spins together (broken down into spin-$\frac{1}{2}$ components). This observation was first outlined in Penrose's binor calculus, which offered a way to decompose spin networks into (the symmetric subspace of) spin-$\frac{1}{2}$ wires meeting at vertices which correspond to their coupling [36]. These binor calculus diagrams can also be viewed as a type of spin network and have previously been converted into a form close to qubit quantum computing via the ZX calculus [38].

At this point, we have an interpretation of coupling spins as relating to the existence of valid triangles with edges determined by the spins involved. However, to see a 'quantised geometry' of triangles, we require both states and operators: the former being mathematical objects from which the latter can meaningfully extract eigenvalues that correspond to geometric properties. For a triangle, these are length and area. Considering two spins coupling to a third, the triangle inequality tells us that if we took the size of the input spins as edges of a triangle, the possible output spins are exactly those that could complete the triangle. A practical and importantly generalisable perspective is to take the three spins as vectors lying dual to the triangle, which we can do since the spin-values obey the triangle inequality, where we note that they will be such that $\sum \vec{j}_i = 0$. When we look at the intertwiner space $\mathrm{Inv}_{SU(2)}\left(\mathcal{H}_{j_1} \otimes \mathcal{H}_{j_2} \otimes \mathcal{H}^*_{j_3}\right)$ where each space corresponds to a spin $j$, we can see this is characterised by a single triangle whose edges lie dual to the spins whose size is dictated by the given spin's magnitudes. The length operator gives us the quantised lengths of the edges of this triangle and is simply the angular momentum operator $\vec{J}$ acting on any of the spins to give $\sqrt{j(j+1)}$. Furthermore for

---

[13]The that we discretise in terms of values of $\frac{1}{2}$ is more a feature of measurement outcomes for spin, the mathematicians index SU(2) by integers without much difficulty.

an area operator we can use $\vec{A} = \frac{1}{2}\vec{j}_1 \wedge \vec{j}_2$ [14]. In line with this, in the case of a three-valent spin network, when intertwiners share an edge, they can be seen as sharing a length of the associated triangle, rendering the entire spin network a tessellating geometry of quantised triangles.

**Quantised tetrahedra**   Let us now consider tetrahedra and proceed in the manner of Ref. [72]. It is a shape composed of 4 triangular faces whose edges are constrained by virtue of coming together to form this shape. It can be usefully characterised by 4 dual vectors $\vec{J}_a, a = 1, \ldots, 4$ lying orthogonal to each face. We say that each $\vec{J}_a = \frac{1}{2}\vec{e}_1 \wedge \vec{e}_2$ where $\vec{e}_i$ are the vectors chosen to represent two of the edges of the triangle whose face lies orthogonal to $\vec{J}_a$. Note how by definition $\vec{J}_a$ lies normal to the faces of the tetrahedra. Let us take these $\vec{J}_a$ to literally be spins, this implies that we have the commutation relation

$$[J^i, J^j] = i\hbar\varepsilon^{ij}{}_k J^k. \tag{C.2}$$

Moreover, as the magnitude of the spins corresponds to the faces, we quantify their area as

$$A = \sqrt{j(j+1)}, \quad j = 0, \frac{1}{2}, 1, \frac{3}{2}, 2, \ldots \tag{C.3}$$

(in general the total angular momentum operator gives the n-1 simplex magnitude of your n-simplex, hence it was length in the triangle case). In this way, every face of the tetrahedra has an area given by their magnitude[15]. One can further show that the following property holds

$$\vec{C} := \sum_{a=1}^{4} \vec{J}_a = 0 \tag{C.4}$$

which is the same condition as seen in the triangular case (again, this persists in higher dimensions). One can also note that the (oriented) volume[16] is given by

$$V^2 = \frac{2}{9}\vec{J}_1 \wedge \vec{J}_2 \wedge \vec{J}_3 = \frac{2}{9}\left(\vec{J}_1 \times \vec{J}_2\right) \cdot \vec{J}_3 = \frac{2}{9}\epsilon_{ijk}J_1^i J_2^j J_3^k = \frac{2}{9}\det J. \tag{C.5}$$

The condition in Eq. (C.4) is crucial because it indicates we can restrict the space in which these quantised tetrahedra live from the Hilbert space $H = H_{j_1} \otimes H_{j_2} \otimes H_{j_3} \otimes H_{j_4}$ to where $\vec{C} = 0$ i.e. $\text{Inv}_{\text{SU(2)}}[H_{j_1} \otimes H_{j_2} \otimes H_{j_3} \otimes H_{j_4}]$. Formally one can show that the closure condition is invariant under the action of an SU(2) rotation [67]. Geometrically, we can get a feeling for this from recalling that SU(2) is essentially SO(3) (i.e., the space of rotations) contracted under the fact that only rays in Hilbert space are physically meaningful. With this in mind, consider that each vector gives the size and position of a triangular face. In general, these vectors could point in any direction, but we are restricted to a tetrahedron. Why is this the case? Well, we can see that in the tetrahedral case, if we move any face relative to the others, then the vectors will no longer sum to zero. They all have to be rotated together, much like rotating the whole tetrahedra. Here, however, we are looking at quantised spins, and so rotations are defined up to rays in Hilbert space, so the rotation group that is really of interest is SU(2). This tells us that our tetrahedral volumes just live in $\text{Inv}_{\text{SU(2)}}[H_{j_1} \otimes H_{j_2} \otimes H_{j_3} \otimes H_{j_4}]$. This principle of invariant volumes tied to vectors summing to zero generalises to arbitrary simplices and tells us that there is a quantised geometric perspective for all dimensions. Interestingly, they can all be reduced back to the three-valent case.

---

[14] $\vec{a} \wedge \vec{b} = \|\vec{a}\|\|\vec{b}\|\sin(\theta)\frac{\vec{n}}{\|\vec{n}\|}$ where $\vec{n}$ is the vector normal to the plane defined by $\vec{a}$ and $\vec{b}$ oriented by the right-hand-rule/cross product convention.

[15] As the vector product of two vectors gives the area of the parallelogram they form, halving this gives that of the triangle.

[16] We have suppressed the natural magnitude units of $\hbar$.

Figure 13: The vertices are the invariant space of four spin-$\frac{1}{2}$s, $\text{Inv}_{\text{SU}(2)}(J_{1/2} \otimes J_{1/2} \otimes J_{1/2} \otimes J_{1/2})$, which written in the form of Eq. (11). In the LQG literature the invariant space of 4 spins is often depicted as a tetrahedron to which this space corresponds when seen in terms of quantised geometry. More conventionally, we can see that this space is spanned by the $J = 0$ and $J = 1$ irrep spaces (which have different dimensions). We also show how these subspaces can be represented as tensor networks corresponding to the two ways to combine the input and output spaces. The triangles correspond to the decomposition of the four-valent vertex into two three-valent spaces, which are viewed as quantised triangles. In our four-valent spin networks circuits, we are directly parameterising these two possible composing triangle geometries for each vertex, which we interpret as a tetrahedra.

**Triangle decomposition** The space $\text{Inv}_{\text{SU}(2)}[H_{j_1} \otimes H_{j_2} \otimes H_{j_3} \otimes H_{j_4}]$ can be broken down into two invariant spaces of three Hilbert spaces. There is some freedom in how they are partitioned but the composite spaces will resemble $\text{Inv}_{\text{SU}(2)}[H_{j_1} \otimes H_{j_2} \otimes H_j^*]$ and $\text{Inv}_{\text{SU}(2)}[H_j \otimes H_{j_3}^* \otimes H_{j_4}^*]$. To see this, we can look to Eq.(A.13), which we can apply twice in this case to give

$$((j_1 j_2)j_3)j_4; jklm\rangle = \sum_{m_1, m_2, m_3, m, n, m_4}^{jm} C^{jm}_{j_1 m_1 j_2 m_2} C^{km}_{jm_1 m_3, m_3} C^{lm}_{k\pi i_4, m_4} \times \bigotimes_{i=1}^4 (j_1, m_i) \qquad \text{(C.6)}$$

where $j \in \{|j_1 - j_2|, ..., j_1 + j_2\}$, $k \in \{|j - j_3|, ..., j + j_3\}$, $l \in \{|k - j_4|, ..., k + j_4\}$, and $n \in \{-l, ..., l\}$, which can be shown to form an orthonormal basis of the space [67]. The crucial part to notice here is that this space is formed by two trivalent spaces with one of the spin spaces summed over (for a more thorough and diagrammatic explanation of this see Ref. [38] or Ref. [67]). The external spins are fixed but the internal space that is summed over points to a particular basis decomposition of the tetrahedra into two pairs of triangles with the different $j$ values at their intersection. For instance let us consider the invariant space of 4 spins. We can deduce that as it is composed of two three-valent invariant spaces, both of which have two components which are spin-$\frac{1}{2}$, they will be decomposed into the case where the internal spin space is $j = 0$ or $j = 1$. Pictorially this is represented in Fig. 13. This principle generalises, and with larger invariant spaces, we get higher order n-simplices (where a triangle is a two-simplex, a tetrahedron a three-simplex, etc) that decompose into $n-1$ triangles with $n-2$ interior edges that give the different possible values which in turn give a possible triangular basis.

## C.3   SU(2) **equivariant algorithms as the search for optimal triangulations**

In short, the geometric approach gives the structure of SU(2) equivariant algorithms a distinctly geometric flavour. Consider that our parameterised spin networks have the specific property that the parameterisation does not alter the input or output space itself, meaning

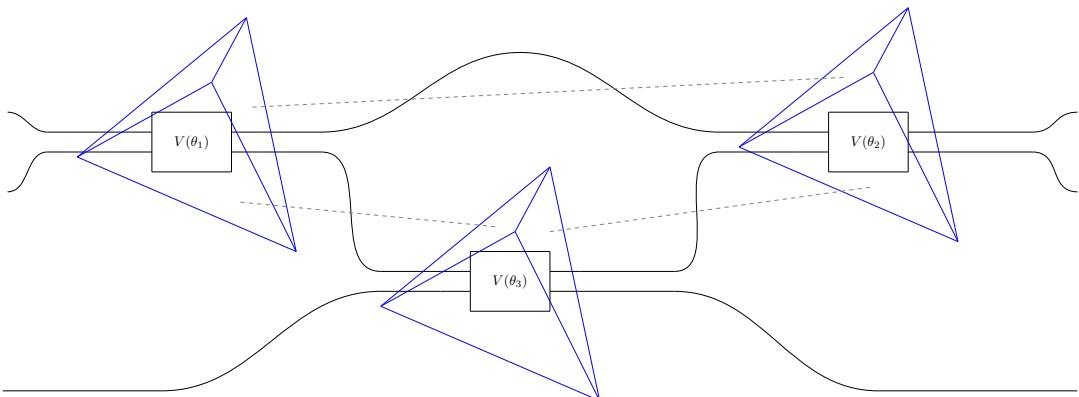

Figure 14: A spin-network circuit with a representation of the three quantised tetrahedra that lie dual to each vertex. Each of their faces has an area of $\frac{\sqrt{3}}{2}\hbar$, which is the total angular momentum of a qubit. The dotted grey lines indicate the faces shared by the tetrahedra that correspond to the qubits passing from the output of one gate to the input of the other. From the perspective of our four-valent spin-network circuits (the two-qubit vertex gates), our variational algorithm is an optimisation of these tessellating tetrahedra (or 5-simplices for the three-qubit vertex gate).

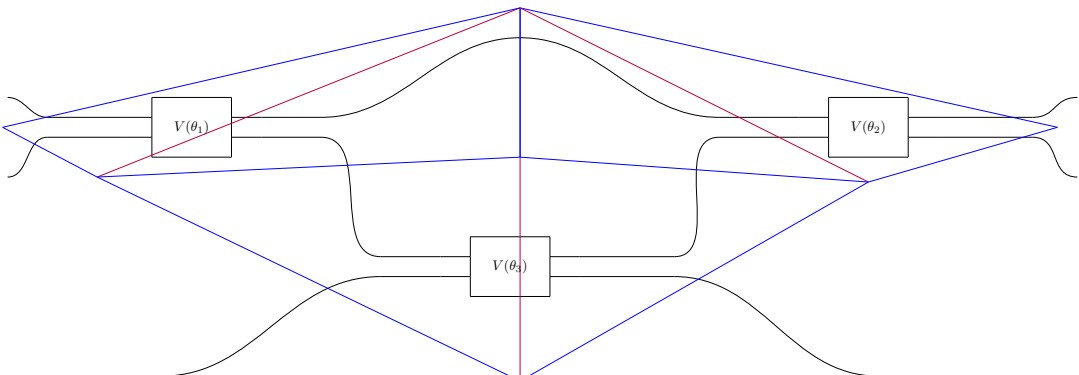

Figure 15: The triangulated interpretation of the spin-network circuit seen as tetrahedra above in Fig. 14. The three tetrahedra have been decomposed into two triangles, each where the exterior edge lengths are fixed at $\frac{\sqrt{3}}{2}\hbar$. The internal red edge, as determined by the intertwiner basis for the tetrahedra, is either 0 or $\sqrt{2}$ which are the eigenvalues for the total angular momentum operator of the internal edge as seen in Eq. (C.3). The phases associated to the different possible measurement lengths for the red edges are the trainable parameters in this network.

that they can be seen as the optimisation of transition maps between $n-1$ simplices (which are the surface of the n-simplex corresponding to the spin-network circuit). In our four-valent example, we can consider the spin networks as quantised tetrahedra or, by flattening this perspective, collections of two triangles whose internal edge lengths correspond to the different internal intertwiner bases elements as seen in Fig. 13.

It is possible to take this geometric perspective further still. We can view our spin networks as maps between quantised 2D spaces in line with Jordan [31]. This presents an interesting perspective of parameterised spin networks as a restricted variety of quantised path integrals. To understand this statement, we should first revisit the concept of spinfoams [72]. In LQG where we have (four-valent) spin networks as a basis of the states of quantised 3D space the spinfoams are the maps between these states. They can be viewed as four simplices whose boundaries are the collections of tetrahedra that make up the initial and final state geometries. They are the discretised equivalent of a particular path in the path integral approach to state transition in that there is a function that acts on them that allows for the computation of amplitude, and the sum over all possible amplitudes gives you the probability of moving from the initial state, i.e. from the faces at one side of the 4-simplex to the final state which are the faces at the other.

This requires that the input and output spaces are fixed in order for it to make sense from the transition amplitude perspective of sending one set of simplices to another, however, as said above, this is exactly what we have for our trainable spin networks. Consider our four-valent spin-network circuits for example, which are formed of tetrahedra and so in this perspective can be viewed as 3D spinfoams. On one side we can see there are the qubits passing into the circuit which can be interpreted as dual to the triangles of tetrahedra. On the other side, the outputs also determine the triangles dual to tetrahedra. We can see then that the specific spin-network circuit is a discretisation of a specific transition path for 2D quantised geometry (because it only uses tetrahedra of a certain size connected in the way specified by the gates, a more general representation would have to remove these restrictions). Looking more broadly at circuits with arbitrary vertex sizes, these amount to collections of simplices of dimensions ranging from 3 to n, with the same restriction that their $n-1$ faces are of size $\sqrt{2}$ (as qubits) and that their connective topology is fixed. These correspond to a more general class of n-degree spinfoams, though one should note that even in the extreme case of one single n-qubit vertex gate that, in principle, runs over every compatible triangle decomposition, it is still premised on a fixed number of internal vertices. The true spin-foam transition amplitude sums over all possibilities, which would include an infinite number of possible internal vertices (naturally, in practice, a normalisation parameter is expected to ensure we arrive at a finite value).

Whatever the order of the $n$-degree of the spinfoam we ultimately use, the optimisation algorithms of our equivariant circuits amount the optimisation of the internal parameterisation of the simplices that make up the transition amplitude. As we have seen above, these can always be decomposed into different tessellations of triangles. Choosing a specific parameter for a vertex gate then amounts to choosing a specific superposition of these internal tessellations with the connective structure of the circuit detailing how these internal tessellations are connected to each other. Though unusual, this is clearly a radically geometric interpretation of SU(2) equivariant algorithms, and it would be interesting to know if this kind of 'geometerisation' generalises to other groups.

## D   Further notes on the Schur gate

**Equivariance of the Schur gate**   Focusing on the Schur matrix in Eq. (13), a natural question is: How are the representations of the group acting on the input affected by the Schur map? As discussed above, the input space has the tensor product representation, and the output has the spin representation, which functions differently. A useful shorthand to express the idea of a group element $g$ acting on some space $H$ without worrying about how exactly it is represented is to write $g \triangleright H$. With this in mind let us consider the action of SU(2) for the two qubit case, for an arbitrary element $g \in \mathrm{SU}(2)$, the input space of the Schur map will transform as $g \triangleright (\mathbb{C}^2 \otimes \mathbb{C}^2) = (g \triangleright \mathbb{C}^2) \otimes (g \triangleright \mathbb{C}^2) = U_g \otimes U_g$, where $U_g$ is the qubit representation of the element $g$. The output space however will transform differently as we are viewing the space as composed of spin components, $g \triangleright (J^0 \oplus J^1) = (g \triangleright J^0) \oplus (g \triangleright J^1) = J^0 \oplus (g \triangleright J^1) = Id \oplus \pi^1(g)$, where we note that the action on the single element spin-0 subspace is trivial and $\pi^1(g)$ is the spin-1 representation of the element $g$. We can use the Schur map itself as a mapping between the tensor product basis and the spin space to create a representation on the direct product, i.e., $U(g)^{\otimes k} = S^\dagger \pi(g) S$, which we can see as mapping our tensor space to the spin-space, performing the group action there, and then sending it back. Let us now see that our Schur map $S$ is equivariant under the action of $g$, which if from a direct and short calculation:

$$S_2(Id \oplus \pi^1(g)) = S_2(S_2^\dagger U(g)^{\otimes k} S_2) = U_g^{\otimes k} S_2. \tag{D.1}$$

The group action has moved from the right-hand side of the Schur gate to the left, and so they commute, which is the definition of equivariance. This calculation, though short and can be somewhat deceptive, it is imperative that we remember that the action of the group should be represented differently before and after the Schur gate. The effect of placing the group action between the Schur gates was to transform it into the appropriate action on the spin space.

A similar discussion applies to the three-qubit space. Recalling that $\mathbb{C}^2 \otimes \mathbb{C}^2 \otimes \mathbb{C}^2 \simeq J^{\frac{1}{2}} \oplus J^{\frac{1}{2}} \oplus J^{\frac{3}{2}}$ we would then say that $g \in G$ acts as $g \triangleright (J^{\frac{1}{2}} \oplus J^{\frac{1}{2}} \oplus J^{\frac{3}{2}}) = (g \triangleright J^{\frac{1}{2}}) \oplus (g \triangleright J^{\frac{1}{2}}) \oplus (g \triangleright J^{\frac{3}{2}})$ and in the end we have that we can use the Schur gate to map us between representations acting on these spaces:

$$S_3(\pi^{\frac{1}{2}}(g) \oplus \pi^{\frac{1}{2}} \oplus \pi^{\frac{3}{2}}(g)) = S_3(S_3^\dagger U(g)^{\otimes k} S_3) = U(g)^{\otimes k} S_3. \tag{D.2}$$

Indeed, this structure will hold in general.

**The Schur gate and PQC recoupling diagrams**   As elements of the spin-basis, the PQC diagrams exactly correspond to the elements of the Schur basis. When specific $J_z$ values are fixed on all the external wires, one can use the PQC diagrams to index the Schur matrix:

$$S_2 = \begin{pmatrix} 1 & 0 & 0 & 0 \\ 0 & \frac{1}{\sqrt{2}} & \frac{1}{\sqrt{2}} & 0 \\ 0 & 0 & 0 & 1 \\ 0 & \frac{1}{\sqrt{2}} & -\frac{1}{\sqrt{2}} & 0 \end{pmatrix} =$$

$$\begin{pmatrix}
c^{1,1}_{\frac12,\frac12,\frac12,\frac12} & c^{1,1}_{\frac12,\frac12,\frac12,-\frac12} & c^{1,1}_{\frac12,-\frac12,\frac12,\frac12} & c^{1,1}_{\frac12,-\frac12,\frac12,-\frac12} \\[4pt]
c^{1,0}_{\frac12,\frac12,\frac12,\frac12} & c^{1,0}_{\frac12,\frac12,\frac12,-\frac12} & c^{1,0}_{\frac12,-\frac12,\frac12,\frac12} & c^{1,0}_{\frac12,-\frac12,\frac12,-\frac12} \\[4pt]
c^{1,-1}_{\frac12,\frac12,\frac12,\frac12} & c^{1,-1}_{\frac12,\frac12,\frac12,-\frac12} & c^{1,-1}_{\frac12,-\frac12,\frac12,\frac12} & c^{1,-1}_{\frac12,-\frac12,\frac12,-\frac12} \\[4pt]
c^{0,0}_{\frac12,\frac12,\frac12,\frac12} & c^{0,0}_{\frac12,\frac12,\frac12,-\frac12} & c^{0,0}_{\frac12,-\frac12,\frac12,\frac12} & c^{0,0}_{\frac12,-\frac12,\frac12,-\frac12}
\end{pmatrix} =$$

$$\begin{pmatrix}
\tfrac12\!\!\succ\!\! 0 \;\; 0;0 & \tfrac12\!\!\succ\!\! 0 \;\; 0;0 & \tfrac12\!\!\succ\!\! 0 \;\; 0;0 & \tfrac12\!\!\succ\!\! 0 \;\; 0;0 \\[6pt]
\tfrac12\!\!\succ\!\! 1 \;\; 1;1 & \tfrac12\!\!\succ\!\! 1 \;\; 1;1 & \tfrac12\!\!\succ\!\! 1 \;\; 1;1 & \tfrac12\!\!\succ\!\! 1 \;\; 1;1 \\[6pt]
-\tfrac12\!\!\succ\!\! 1 \;\; 1;0 & -\tfrac12\!\!\succ\!\! 1 \;\; 1;0 & -\tfrac12\!\!\succ\!\! 1 \;\; 1;0 & -\tfrac12\!\!\succ\!\! 1 \;\; 1;0 \\[6pt]
-\tfrac12\!\!\succ\!\! 1 \;\; 1;-1 & -\tfrac12\!\!\succ\!\! 1 \;\; 1;-1 & -\tfrac12\!\!\succ\!\! 1 \;\; 1;-1 & -\tfrac12\!\!\succ\!\! 1 \;\; 1;-1
\end{pmatrix}$$

$$\tag{D.3}$$

In the final equality, we write the diagrams as the corresponding matrix with total $J$ values written above the wires and the $J;J_z$ values written horizontally to them.

The connection becomes clearer in the three-qubit case, showing how the entries of the matrices are the combinations of Clebsch-Gordan coefficients that correspond to particular coupling structures:

$$S_3 = (c^{j_4,m_4}_{j_1,m_1;j_2,m_2}\, c^{J,M}_{j_4,m_4;j_3,m_3}) = \begin{pmatrix}
1 & 0 & 0 & 0 & 0 & 0 & 0 & 0 \\
0 & \frac{1}{\sqrt3} & \frac{1}{\sqrt3} & 0 & \frac{1}{\sqrt3} & 0 & 0 & 0 \\
0 & 0 & 0 & \frac{1}{\sqrt3} & 0 & \frac{1}{\sqrt3} & \frac{1}{\sqrt3} & 0 \\
0 & 0 & 0 & 0 & 0 & 0 & 0 & 1 \\
0 & \sqrt{\frac23} & -\frac{1}{\sqrt6} & 0 & -\frac{1}{\sqrt6} & 0 & 0 & 0 \\
0 & 0 & 0 & -\frac{1}{\sqrt6} & 0 & -\frac{1}{\sqrt6} & \sqrt{\frac23} & 0 \\
0 & 0 & \frac{1}{\sqrt2} & 0 & \frac{1}{\sqrt2} & 0 & 0 & 0 \\
0 & 0 & 0 & -\frac{1}{\sqrt2} & 0 & \frac{1}{\sqrt2} & 0 & 0
\end{pmatrix} \Leftrightarrow$$

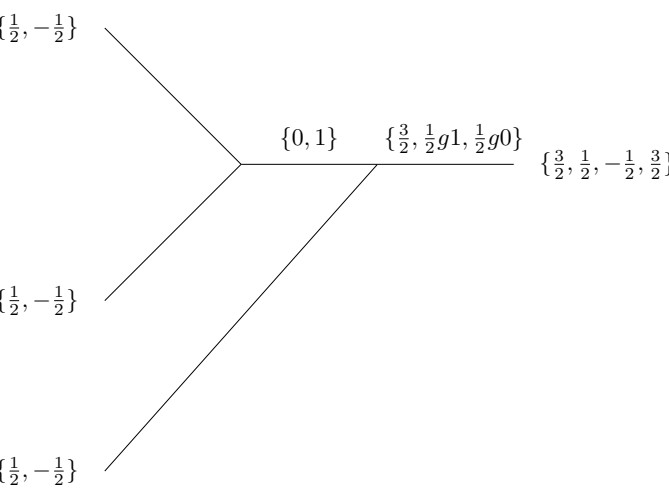

For reasons of space, we merely outline a single diagram with the possible indices highlighted (which is why we don't use equality with the last line). The terms $\frac12 g1$ and $\frac12 g0$ serve to

separate the two ways one can couple to a total angular momentum of $\frac{1}{2}$ on the last edge. Specifically, $\frac{1}{2}g1$ indicates the case when the initial coupling resulted in a total angular momentum of 1, and $\frac{1}{2}g0$ is for when it resulted in 0. These have to be distinguished as they correspond to the multiplicities of spin-$\frac{1}{2}$ and so do actually index different elements in the matrix. Here, we merely state the $J_z$ values at the sides of the wires on the RHS, and we assume the $J_z$ values range only where permissible.

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
