# Peer review of "All you need is spin: SU(2) equivariant variational quantum circuits based on spin networks"

_SciPost Physics_

## Round 1 · Referee Report · Anonymous · 2024-3-18

Strengths

1) Well-written and very pedagogical
2) Clarity of the results

Weaknesses

1) Lack of comparison with other methods/discussion of complexity scaling
2) Lack of original content (or lack of visibility of what is truly original)

Report

Summary
This article proposes to use SU(2) symmetric (SU(2) equivariant) quantum circuits to describe the groundstates of SU(2) Hamiltonians on a quantum computer, using the variational quantum eigensolver algorithm to optimize the circuit parameters.
The authors first provide a detailed introduction to SU(2) invariant tensor networks/quantum circuits and their representation. They then present an equivalence between permutation gates and the SU(2) equivariant unitary gates. Finally, they benchmark their algorithm on the one-dimensional J1-J2 Heisenberg chain and on the antiferromagnetic Kagome system (through classical simulations).

Report
First, I would like to praise the authors for the quality of their pedagogical introduction. Generally, the paper is very well-written and easy to follow. Nonetheless, I do not think I can recommend the publication of the paper in SciPost Physics in its current form.

Main objection
My main objection for publication is that I do not believe that this paper provides a significant enough breakthrough to deserve publication in SciPost Physics.

Secs 2 and 3 are mostly introductory/paraphrase previous results (note that the authors are honest about this fact). It is fairly straightforward SU(2) representation theory applied to tensor networks, the only caveat being that the tensors are unitary.
Sec. 4 deals with the relation between irreps of the symmetric group and SU(2) irreps. While interesting, it appears to me to only be a marginal generalization of the results of Ref. 20 and 22.
Sec. 5 does present original numerical results, but only as a short benchmark (see also questions below) and without attempting to solve any outstanding issue (understandably given the proposed algorithms are for a quantum computer).
Sec. 6 is an interesting but purely conceptual discussion.

Questions and comments
Methods
1) What is the difference in representability between unitary quantum circuits with finite depth/range and more standard tensor networks such as PEPS? I understand the speed up for a quantum computer.

2) In your algorithm, you use up to 3 vertex gates (due to complexity of physical implementation?). How would you expect your results to change if you were able to directly optimize a gate over the hexagonal cell in your kagome lattice?
If I am not mistaken, two sites gate are not enough to describe all possible circuits. Can you comment on that and your use of 2 tensors for the 1d model?

3) What is the fundamental difference between your circuit and the one used in Ref. 60 for example, beyond your form being more systematic and generic?

Numerical results
1) There are no comparisons with classical methods and no discussions of the scaling of the number of circuit parameters with the system size. A reminder (just a formula) of the complexity of the vQE per sweep would be appreciated. A discussion/numerical evidence of the required number of sweeps (and its scaling with system/parameter) and of the barren plateaus would also be needed in general to properly assess the efficiency of the representation.

2) The one-dimensional Heisenberg system is exactly solvable for $J_2 = 0$ (by Bethe Ansatz) and J2 = 0.5 (Majumdar Gosh point). In particular, $J_2 = 0.44$ is close to the MG point which can be written as a product of independent singlets. The entanglement content of the MG state is therefore very low, and it is representable by an exact matrix product states. Correspondingly, at $J_2 = 0.44$, the groundstate is gapped and has a very short correlation length.
As such, it is surprising that your algorithm struggles more to converge for $J_2 = 0.44$ than for $ J_2 = 0$ (gapless). Could you comment on that?
Additionally, the energy appears to saturate with the number of parameters (number of layers) before jumping closer to the exact value and with a large spread for the deepest circuit. Could you explain why? Is it related to barren plateaus/an absence of solutions?

3) The Kagome antiferromagnet is a notoriously difficult problem, with very small gaps in finite systems (see e.g. https://journals.aps.org/prb/abstract/10.1103/PhysRevB.100.155142 for a not so recent ED work). As such, a discussion on the scaling with system sizes and comparison with existing methods is sorely lacking here. I have also similar concerns as in 2) given the spread of the obtained energy, and the previous observed saturation in the 1d model.

Requested changes

Major changes
- Clarify the original contribution of the paper
- If the numerical results are to be important, proper discussions on scalings and barren plateaus are needed, as well as comparison with other techniques.

Minor changes
- Some notations could be clarified (the Schur gate and the symmetric group are denoted by $S_n$, the generator of the SU(2) equivariant gates is in one section H and in the following $\mathcal{T}$, etc).
- Better bibliography on the physics side of the chosen benchmark problems
- Minor typo p.19. Above 4.3 "exponetiated"

---

## Round 1 · Referee Report · Anonymous · 2024-5-6

Strengths

1. Overall, the paper is nicely written
2. The authors made an effort to make it as self-contained as possible by providing a lot of background content*
3. The topic is very relevant as poor ansatz design leads to scalability issues in VQE
4. Although this is not my area of expertise the relevant literature seems to have been cited as numerous papers related to equivariant circuits are referred to

Weaknesses

1. Lack of clear message about the scope of the method and how it compares to other ansätze
2. Overly technical without proper justification being brought for the level of details provided*
3. Many subjects are referred to for further study, but only very superficially
4. I am still wondering who is the target audience for a paper solving condensed matter Hamiltonians within VQE but making connections with loop quantum gravity

*a strong textbook reference about the representation theory of $SU(2)$ could have helped achieved a more trimmed paper and left space for a pedagogical view

Report

Foreword: This review reflects the point of view of someone having expertise in variational circuits and their use to solve condensed matter problems. Conversely the author of this review does not hold any expertise in spin networks.

The paper provides a method to construct circuits which are equivariant with regards to the $SU(2)$ symmetry, based on the representation theory of $SU(2)$. It is stated that
(a) unlike previous approaches, theirs is simpler to implement on quantum hardware, and
(b) that VQE methods are boosted by their ansatz design strategies.

The paper is structured as follows. A general introduction is followed by a section providing background on the theory of spin networks. Then the method to build equivariant gates based on the representation theory of SU(2) is presented in section 3. Section 4 is technical and establish the equivalence between SU(2) equivariant gates and generalised permutations and that twirling all generators of the unitary group enables to recover the whole set of equivariant gates generators. The next section presents simulation results for the Heisenberg model on the 1d-triangular and on the Kagome lattices. The results are discussed in section 6, which also suggests connections with other theories. Section 7 concludes.

Despite the fact that I think there is some content of interest for the community, which could correspond to Expectation 1 of this journal, I cannot recommend the publication of the paper in SciPost in its current form. I have identified two major caveats, both on the substance and on the form of the article. The first concern resides in the absence of benchmark against other approaches - and, resultingly, the absence of a clear message. The claims presented in the introduction are vague and not backed by evidence. The second is about the readability of the article, which is both content-heavy and sometimes too superficial for the points the authors are trying to make to be conveyed efficiently. This lead me to consider general acceptance criteria 4 and 5 not to be met. I expand on these two concerns below.

1. In the introduction the motivation is given in a QML framework (and in particular, in the context of geometric QML where equivariant circuit structures were explored in previous work) whereas the results are concerned with a physical model. The reasoning behind this choice is not given, if not by the fuzzy statement 'While our circuits can be used for usual machine learning tasks [...] we choose the problem of finding the ground state of $SU(2)$ symmetric Hamiltonians as it provides a better benchmark platform for classically simulated QML models (with ~ 20 qubits).' on p3. I guess the authors meant that this benchmark was amenable to classical simulation, unlike 'proper' ML benchmarks (e.g. classification) which require to simulate a prohibitive number of qubits.

However, it is of paramount importance that the scope of this work be well defined. Applying geometric arguments to design symmetry-preserving ansatz circuits for the ground state problem comes with different challenges compared with classification tasks in ML, as underlined in Ref [24] cited by the authors. Indeed in [24] it is stated that equivariance comes with a generalization/expressibility trade-off: symmetry has been shown to yield enhanced generalization capabilities in ML due to the overfitting being avoided. On the other hand, the gain for the ground state problem tackled within VQE lies in a reduced parameter count once sufficient expressibility is recovered by considering enough layers in the ansatz. As a consequence I would expect the authors to compare equivariant architectures to non-equivariant ones to show that their technique is indeed associated with a lower parameter count for a given (and sufficiently high) target accuracy in the estimation of the ground state energy. Indeed in Ref [24] the authors apply the twirling technique to construct an equivariant version of the QAOA ansatz for a Heisenberg chain of 10 spins and find that the equivariance property is associated with better accuracies once a certain depth is reached. Moreover, if reducing the parameter count for better trainability is indeed the scope of this work, literature for condensed matter should be cited in introduction: LDCA circuit, UCCSD, HVA.

One of my main pain points in that the numerical study proposed by the author does not meet this prospective scope, and neither has a clear aim.

First they consider a Heisenberg Hamiltonian with 20 sites arranged in a 1d-triangular lattice. They choose to consider $J1=1$ and either $J2=0$ or $J2=0.44$, arguing for the latter $J2$ value that it corresponds to a maximally-challenging setting for some classical methods, albeit still tractable within DMRG owing to the 1d nature of the problem. The selected case could be considered as a testbed with an 'easy' problem instance $J2=0$ and a 'difficult' one $J2 \neq 0$ to compare their ansatz to other ansätze (such as those cited above, or other symmetry-preserving ansatz, e.g. https://www.nature.com/articles/s41534-019-0240-1), with and within the context of geometric ML. Surprisingly, the authors chose instead to compare the two-qubit vertex and the three-qubit vertex versions of their ansatz. They plot a measure of the inaccuracy in the attained GS energy as a function of the number of parameters for each version of their ansatz for 16 different initializations. Their main selling point is that the accuracy increases with the depth for the 3-qubit vertex version whereas the 2-qubit version saturates for the $J2=0.44$ problem instance. The authors present this result as surprising due to the Hamiltonian having only up to two-body interaction terms: I am not fully convinced this is surprising, as it could boil down to assigning the same parameter to two different two-qubit gates and thus enforcing another symmetry.

In the second study, concerned with the kagome lattice, only the three-qubit vertex ansatz is studied. The choice not to consider the two-qubit vertex ansatz anymore is not justified in the text. This time however the authors make a very welcome comparison with documented accuracies within VQE. It is unfortunate that the comparison is made only in the text and not included as a reference line on the plot of Fig 9.

Considering the above, I do not deem the claims stated in the introduction to be backed. The 'boosting' of the VQE procedure (claim (b) ), whatever this term means, is not assessed. Parameter counts should be compared. As for claim (a) (easy implementation), it seems to be contradicted by the authors p14: 'However, it is generally difficult to decompose a spin-network circuit with arbitrary θ⃗ to single and two-qubit parameterised quantum gates with a fixed structure and so this is a compilation task that requires further study (i.e., finding a circuit with single and two-qubit parameterised gates that generate the equivariant gate).' In that regard, it is worth-noting that there exists a trade-off between the gate count and the parameter count (for instance, within ADAPT-VQE, different operator pools can be selected to favour one criterion over the other), so that in spite of a gentle scaling gains in terms of parameter count could be counterbalanced by a prohibitive gate count.

2. I have found the article difficult to read and never reached a conclusion as to what the authors were trying to achieve. For instance the authors hint at a diagrammatic approach based on spin networks to design ansatz circuits, but that approach is never fully expanded and contrasts with the simplistic form of their ansatz (the long-awaited formulae 74 and 75). Many concepts are introduced, including a full section (section 4) which the authors warn that it can be skipped. I wondered why this section was not part of the appendix. Connections are made with permutational quantum computing as well as loop quantum gravity (the latter being introduced through a half a page paragraph very optimistically titled 'What is loop quantum gravity?'), but these connections are only alluded to. As a result, the paper sometimes looked to me more like a mix between a compilation of notes from the authors and a roadmap than a properly finalised scientific article.

On a side note:
- For 400 parameters the 3-qubit gates ansatz yields extremely spread results, partly accumulating around the accuracy of the 2-qubit ansatz and partly providing half a decade of increase in the accuracy compared with the previous depth. In my opinion this spread should be commented on.
- The results plots are quite hard to read so I'd suggest to plot additionally the mean accuracy with error bars reflecting the variance.
- Two hyperparameters of the VQE optimization are not specified (the rescaling factor $\alpha$ and the learning rate of the Adam optimizer, although these can be read in the code)
- The authors should by all means try to avoid unscientific statements such as 'The Clebsch-Gordan conditions are more interesting than they appear. '(p. 42)
- Definition of the 3-qubit Schur gate in (16) does not match that of the code (there is a reshuffling of the rows but also different relative signs at some location)
- Mispellings 'triangluar' p23, (one) 'ansätze' instead of ansatz on various occasions
- When citing references which follow each other, the authors use the formatting '10-11-12-13-14'. The readability would be improved by writing this as '10-14'.

Requested changes

I believe the authors could be provided a valuable input to the research literature by
1. Only providing carefully-picked technical details with an emphasis on pedagogy
2. Benchmarking against other ansätze
3. Clarifying and backing claims with evidence
4. Leaving connections with other fields such as loop quantum gravity to further work

Recommendation

Reject

---

## Editorial Decision

awaiting_resubmission